# Toxic Effects and Mechanisms of Polybrominated Diphenyl Ethers

**DOI:** 10.3390/ijms241713487

**Published:** 2023-08-30

**Authors:** Jinsong Xue, Qingqing Xiao, Min Zhang, Dan Li, Xiaofei Wang

**Affiliations:** School of Biology, Food and Environment, Hefei University, Hefei 230601, China; xiaoqq@hfuu.edu.cn (Q.X.); zhangmo@hfuu.edu.cn (M.Z.); lidan@hfuu.edu.cn (D.L.)

**Keywords:** polybrominated diphenyl ethers, exposures, toxic effects, toxic mechanisms

## Abstract

Polybrominated diphenyl ethers (PBDEs) are a group of flame retardants used in plastics, textiles, polyurethane foam, and other materials. They contain two halogenated aromatic rings bonded by an ester bond and are classified according to the number and position of bromine atoms. Due to their widespread use, PBDEs have been detected in soil, air, water, dust, and animal tissues. Besides, PBDEs have been found in various tissues, including liver, kidney, adipose, brain, breast milk and plasma. The continued accumulation of PBDEs has raised concerns about their potential toxicity, including hepatotoxicity, kidney toxicity, gut toxicity, thyroid toxicity, embryotoxicity, reproductive toxicity, neurotoxicity, and immunotoxicity. Previous studies have suggested that there may be various mechanisms contributing to PBDEs toxicity. The present study aimed to outline PBDEs’ toxic effects and mechanisms on different organ systems. Given PBDEs’ bioaccumulation and adverse impacts on human health and other living organisms, we summarize PBDEs’ effects and potential toxicity mechanisms and tend to broaden the horizons to facilitate the design of new prevention strategies for PBDEs-induced toxicity.

## 1. Introduction

Polybrominated diphenyl ethers (PBDEs) can inhibit combustion in organic material and suppress toxic fumes formation. They are thus found in various products, including electronics, vehicles, plastics, furnishings, polyurethane foams, building materials and textiles [1]. As halogenated organic compounds, PBDEs consist of two benzene rings connected by an oxygen atom. A total of 209 PBDE congeners named according to the number of bromine atoms and their position (i.e., ortho-, meta-, and para-substitution) are included in PBDEs [2,3]. Among these congeners, the major components available for commercial use are pentabromodiphenyl ether (penta-BDE), octabromodiphenyl ether (octa-BDE), and decabromodiphenyl ether (deca-BDE, PBDE-209) [4]. However, despite their efficiency in preventing fires, extensive use of PBDEs has engendered great safety concerns for the environment and public health [5]. Therefore, the present study aims to summarize the hazardous effects and potential mechanisms of PBDEs.

Since PBDEs are not chemically bonded to the polymer product, they are easily released into surroundings and become persistent organic pollutants, leading to contamination of the external environment [6]. PBDEs are distributed throughout the world and are frequently found in air, soil, water, and biota. For example, PBDEs enter the environment through atmospheric emissions from various sources, such as waste incineration, manufacturing, and recycling infrastructures [7]. The generated pollutants are transported long distances via airflow, resulting in deposition and accumulation in distant regions [8]. In addition, disposal of PBDE treatment materials in landfill sites and illegal sites may lead to the emission of leachate, thereby imposing major soil pollution issues [9]. Owning to the lipophilic and hydrophobic properties, PBDEs bind firmly to organic matter and remain in soils with reported half-lives of about 28 years [10]. The contaminated soils may also transfer PBDEs to suspended solids and sediments of aquatic environments via precipitation run-off [11]. Water is an important medium for PBDEs transmission [12]. Therefore, a persistent concern has been raised about the increasing levels of PBDEs in sewage treatment works [9,13]. PBDEs can enter the human body via ingestion of dust-bound PBDEs and inhalation of air-containing PBDEs [14]. In addition to the direct exposure, PBDEs are absorbed by the plants’ roots and shoots that eventually enter the food chain [15]. PBDEs in fish, meat and livestock products may also result in a dietary risk and increase the PBDEs body burden in human beings [16,17].

People are unwittingly exposed to chemicals through their food, drinking water, the air they breathe, dust, and contact with consumer goods. Therefore, PBDEs may affect overall health by interacting with other substances. For instance, when microplastics and PBDEs were both present, the oxidative system was more severely disrupted than when either was present alone [18]. Interestingly, those who opt for the high-fat diet (HFD) are more at risk of BDE-209, exacerbating the advancement of non-alcoholic fatty liver disease (NAFLD) [19]. The combination of cadmium and PBDE-209 exposure resulted in more severe damage to hepatocytes [20]. Telomere length among newborns is linked to prenatal exposure to mixtures of per- and polyfuoroalkyl substances (PFAS) and PBDEs [21]. Besides, prenatal PBDEs can adversely affect child health, so the exposure of pregnant women to PBDEs cannot be ignored. For example, it was determined that prenatal exposures to PBDEs correlated with heightened liver injury risks, impaired cognitive performance, and fetal growth restriction in children [22,23,24]. In this review, we summarize the literature regarding the effects of combined and indirect exposures, which could further clarify PBDEs’ detrimental effects.

PBDEs are ubiquitous toxicants frequently detected in human tissues [25]. As an essential organ for the metabolism of exogenous compounds, the liver serves as the main target for PBDEs [26,27]. For instance, the transplacental transfer of PBDEs from mother to fetus leads to tissue accumulation in the fetal liver [28]. Various PBDE congeners, such as PBDE-3, PBDE-7, PBDE-17, PBDE-99, PBDE-153, PBDE-197, and PBDE-209, have been reported to accumulate in blood, hair, and nails [29,30]. Hair and nail samples are non-invasive biomonitors. PBDE-47 and PBDE-99 were the predominant PBDEs detected in hair and nail samples [31]. Moreover, PBDEs have been found in human breast milk, cord blood and placentas. Therefore, it’s inevitable that newborns are exposed to high levels of PBDEs during prenatal and postnatal periods [32]. Additionally, PBDEs were reported in adipose, kidney, lung, and semen [33].

Because of concerns regarding PBDEs’ persistence, bioaccumulation, and potential toxicity, numerous studies have focused on delineating the underlying mechanisms. Herein, we review the literature addressing the effects on different tissues and mechanisms known to potentially contribute to PBDEs toxicity. The summarized information of this study may provide a clearer understanding of the impact of PBDEs on health.

## 2. Liver Toxicity

The liver is an essential organ for metabolic detoxification and is sensitive to environmental toxicants. Hence, the liver is susceptible to injury when exposed to xenobiotics [26,34,35]. For instance, significant liver weight increase and cell swelling, coupled with an elevated expression of cytochrome P450 (CYP1A2, CYP3A1, and CYP2B1) enzymes and genes and hepatocytic fatty degeneration, have been reported in PBDEs (the structures are shown in Figure 1) treated animals [36,37,38,39,40,41].

### 2.1. Oxidative Damage and Apoptosis

A study on zebrafish has shown that PBDE-47 and PBDE-153 exposure markedly increased catalase (CAT) and superoxide dismutase (SOD) activities [26]. Additionally, the upregulation of apoptotic-regulated genes, including cysteine-aspartic acid protease-3 (*Caspase-3*) and tumor protein 53 (*P53*), as well as downregulation of anti-apoptotic genes, including B-cell lymphoma 2 (*Bcl2*) were observed in zebrafish treatment with PBDE-47 and PBDE-153, indicating PBDEs may regulate hepatic oxidative stress, DNA damage and apoptosis [26]. In addition, PBDE-47 and PBDE-32 reduced cell viability, generated reactive oxygen species (ROS) and triggered apoptosis in human hepatocellular carcinoma cell line HepG2 cells [42,43]. Shao et al. have analyzed the response of primary human fetal liver hematopoietic stem cells (HSCs) to PBDE-47 induction. They found higher concentrations of PBDE-47 may elicit overt ROS generation and lipid peroxidation, whereas N-acetylcysteine (NAC) can alleviate oxidative damage induced by PBDE-47 [44]. Analogously, trout liver cells exposed to PBDE-47 displayed a significant reduction in cell viability. The enhanced 6-carboxy-2′,7′-dichlorodihydrofluorescein diacetate (H2DCFDA) fluorescence in the presence of PBDE-47 indicated liver cells may be sensitive to PBDE-47 via a mechanism involving oxidative stress [45]. Zhang et al. investigated a rescue strategy using troxerutin to ameliorate PBDE-47-induced hepatocyte apoptosis. Perturbation of proteasome functions leads to endoplasmic reticulum (ER) stress, which is associated with apoptosis. They found that troxerutin efficaciously mitigates mice’s liver apoptosis via modulating oxidative stress-mediated proteasome dysfunction. Furthermore, the downstream TNF receptor-associated factor 2 (TRAF2)/apoptosis signal-regulating kinase 1 (ASK1)/c-Jun N-terminal kinase (JNK) pathway was dramatically blocked by troxerutin in PBDE-47-treated mice livers [46]. Meanwhile, PBDE-47 promotes liver inflammation by inducing oxidative stress-triggered nicotinamide adenine dinucleotide (NAD^+^) depletion. Troxerutin may abate oxidative stress, preventing the NAD^+^-depletion-mediated loss of silent mating type information regulation 2 homolog 1 (Sirt1) and subsequent occurrence of inflammation [47]. In rat liver, PBDE-99 induced oxidative damage as evidenced by increased SOD activity and oxidized glutathione (GSSG) level, as well as decreased glutathione (GSH) level and CAT activity [48]. Likewise, PBDE-99 activated Caspases (i.e., Caspase-3 and Caspase-9) and generated toxic levels of ROS, thereby causing HepG2 cell apoptosis [49]. PBDE-209 and its quinone-type metabolite could induce an oxidative stress response, which activates ER stress and the autophagy-lysosomal system in hepatocytes [50,51]. Meanwhile, PBDE-209 disrupted calcium homeostasis, induced mitochondrial Ca^2+^ overload, and the subsequent cell apoptosis occurred [50,52]. Hu et al. have conducted several experiments to assess oxidative stress indicators. For example, increased ROS and lactate dehydrogenase (LDH) leakage have been observed in HepG2 cells dosed with PBDE-209 [53]. PBDE-209 could upregulate the activity of hepatic glutathione reductase (GR), and this elevation may compensate for cellular GSH depletion [54]. Interestingly, in 2013, samples of the kingfisher were collected from the e-waste recycling site and processed for biochemical analysis. The analysis showed that PBDEs, malondialdehyde (MDA) and ROS levels in kingfishers from e-waste sites were markedly increased compared with the normal group. Conversely, SOD and CAT activities in the liver from the exposed area were lower than in the reference group [55]. Transcriptional profiles of *O.melastigma* liver were analysed. The results discovered that PBDE-47 may activate phosphoinositide-3-kinase (PI3K) and mitogen-activated protein kinase (MAPK) pathway, which can modulate cell growth, proliferation, and survival [56]. The mechanisms are shown in Figure 2 and Figure 3.

### 2.2. Disturbance of Glucose and Lipid Metabolism

A growing body of evidence supports the idea that exposure to PBDEs is associated with metabolic dysfunction, with findings suggesting that these toxins may interfere with glucose and lipid metabolism. PBDE-47 and PBDE-153 have been reported to alter the blood-liver balance of lipids and disturb glucose metabolism in mice [1,57]. Moreover, to test if the aberrant metabolic phenotype is associated with altered liver epigenome, adult rats were exposed to PBDE-47, and functional analysis displayed that genes related to differentially methylated regions and differentially expressed miRNA were involved in lipid metabolism [58]. PBDE-71 has been found to reduce the activity of phosphoenolpyruvate carboxykinase, a key metabolic enzyme in hepatic glucose and lipid metabolism, and change the glucose: insulin ratio [59,60]. C57BL/6 mice that received PBDE-71 exhibited glucose intolerance, fasting hyperglycemia, retarded glucose clearance, and diminished thermogenic brown adipose tissue mass [61]. Zhu et al. reported PBDE-209 altered protein kinase A (PKA), phospho-PKA (p-PKA), adenosine 5′-monophosphate-activated protein kinase (AMPK), phospho-AMPK (p-AMPK), acetyl-CoA carboxylase (ACC), and fatty acid synthase (FAS) expression in rats’ liver and LO2 cells (human normal liver cells). Besides, protein kinase cyclic adenosine monophosphate (cAMP)-activated catalytic subunit α (*PRKACA-1*) hypermethylation induced by PBDE-209 was observed in LO2 cells. Further study revealed that hypermethylation may contribute to disturbance of glycolipid metabolism [62]. Casella et al. exposed HepG2 cells to PBDEs (i.e., PBDE-47, PBDE-99 and PBDE-209) at 1 nM. The following Kyoto Encyclopedia of Genes and Genomes (KEGG) pathways and Gene Set Enrichment Analysis (GSEA) analyses were carried out, and the results indicated that PBDE-47 perturbed the glucose metabolism and hypoxia pathway; the ternary mixtures containing PBDE-47, PBDE-99 and PBDE-209 influenced lipid metabolism and PI3K/protein kinase B (AKT)/mammalian target of rapamycin (mTOR) signaling pathway. Meanwhile, PBDE-209 was reported to cause increased estrogen receptor *α* (*ERα*) and peroxisome proliferator-activated receptor α (*PPARα*) gene expression. These mechanism-based findings may reveal the potential relation between PBDEs and glycolipid metabolism [27]. PPARγ is an important nuclear receptor crucial in regulating lipid metabolism and glucose homeostasis [63]. Of interest, PBDE-47, a potential PPARγ ligand, could activate PPARγ [64,65]. PPARγ may push the adipocyte differentiation process forward by forming a positive-feedback loop with liver X receptor α (LXRα) [66]. It was reported that PPARγ activated by PBDE-47 may increase the expression of adipocyte-specific genes such as fatty acid binding protein 4 (*Fabp4*), lipoprotein lipase (*Lpl*), glucose transporter type 4 (*Slc2a4*), and adiponectin (*Adipoq*) [67]. Zhu et al. have found PBDE-209 led to histological impairment and lipid deposition, which were characterized by reduced glycogen and high-density lipoprotein (HDL) levels and increased low-density lipoprotein (LDL), glucose, triglyceride (TG) levels, and total cholesterol (CHOL) in mice livers. And besides they also found that LO2 cells’ survival declined after PBDE-209 treatment. Further exploration revealed that PBDE-209 impaired glucose homeostasis via preventing PI3K/AKT/Glucose transporter type 4 (GLUT4) signaling pathway and induced lipid metabolic abnormality by triggering mTOR/PPARγ/retinoid X receptor α (RXRα) signaling pathways [68,69]. The mTOR pathway activated by PBDE-209 is responsible for the induction of PPARγ expression. Subsequently, PPARγ increases lipogenesis by combining with RXRα to form dimers [68]. Intriguingly, PPARγ inhibitor antagonized the alterations to the expression of p-mTOR, PPARγ, and RXRα and hindered TG accumulation provoked by PBDE-209, suggesting PPARγ may participate in modulating glucolipid metabolism [68]. Rats orally administered with PBDE-209 have shown hyperglycemia as compared to control rats. The reduced GSH and SOD implied that oxidative damage may contribute to PBDE-209-induced hyperglycemia and the onset of diabetes [70]. PBDE-209 has been reported to hinder glucose absorption, increase the levels of total cholesterol (TC), TG, aspartate transaminase (AST), alanine aminotransferase (ALT), and MDA through insulin receptor substrate-1 (IRS-1)/GLUT4 and IRS-1/PI3K/AKT/Glycogen synthase kinase 3β (GSK-3β) pathways, eventually interfering with glucolipid metabolism in buffalo rat liver cells with insulin resistance (IR-BRL) [71]. The mechanisms are shown in Figure 4.

### 2.3. Mitochondrial Damage

PBDE-47 increased *miR-34a-5p* level to trigger NAD^+^ insufficiency via targeting nicotinamide phosphoribosyltransferase (NAMPT) expression. Subsequently, Sirtuin 3 (Sirt3)/forkhead box O-3 α (FOXO3α)/PTEN-induced putative kinase1 (PINK1) pathway-associated mitophagy was inhibited, which results in mitochondrial dysfunction and oxidative damages in mouse livers [72]. Fetal liver HSCs with PBDE-47 treatment showed a loss of mitochondrial membrane potential (MMP) [44]. DNA damage and mitochondrial impairment were detectable in cells after exposure to PBDE-47 and PBDE-32 [42]. Pazin et al. have found that PBDE-47 or PBDE-99 can influence membrane potential, mitochondrial inner membrane, oxygen consumption, mitochondrial swelling, and calcium release, which results in adenosine triphosphate (ATP) exhaustion [73]. As the energy-producing organelles inside cells, mitochondria are essential in maintaining energy supplies. In isolated liver mitochondria, Pereira et al. observed that PBDE-153 can interact with the mitochondrial membrane and disrupt MMP, thus causing ATP deficiency [74]. Meanwhile, they have also investigated the effects of PBDE-209 on rat liver mitochondria. The results showed PBDE-209-induced matrix swelling and ATP depletion. This process may contribute to reduced HepG2 cell viability [75]. The mechanisms are shown in Figure 5.

### 2.4. Indirect Exposures

Indirect exposures occurred perinatally. For example, Dunnick et al. reported that PBDE-47 induced centrilobular hypertrophy and fatty change in pup livers on postnatal day (PND) 22. Liver transcriptomic changes were also measured, and the results showed that cytochrome p450 transcripts, nuclear factor erythroid 2-related factor 2 (Nrf2) antioxidant pathway transcripts and ATP-binding cassette (ABC) membrane transport transcripts were upregulated. These alterations elicited lipids, oncogenes, and epigenetic changes, which can lead to liver damage and tumorigenesis [76]. Perinatal exposure to PBDE-99 can disrupt the nongenomic actions of thyroid hormone (TH), thereby reducing the activity of the PI3K/AKT pathway in rat pup livers and affecting cell survival [77].

### 2.5. Combined Exposures

Combined exposures produced a series of public health issues. It has been reported that PBDEs are tightly associated with the occurrence of obesity and NAFLD. Further exploration revealed that the combination of PBDE-47 and HFD treatment reduced carnitine palmitoyltransferase 1α (*CPT1α*) gene expression, inhibiting fatty acid oxidation. Besides, the expression of microsomal TG transfer protein was inhibited by PBDE-47, which led to dysfunction of TG metabolism [78]. Co-exposure of nanoplastics and PBDE-47 leads to changed liver colour and atrophied liver in zebrafish larvae. The liver degeneration or necrosis may be associated with reduced antioxidant glutathione peroxidase 1 (*gpx1a*) gene and increased *CYP1A1* [79]. Using high-throughput sequencing approaches, Li et al. have proved that combined exposure of microplastics and PBDE-47 upregulated PPAR-related genes and reduced IL-17-associated genes [18]. Chen et al. have found that combined exposure to PBDE-209 and high fat resulted in elevated TG, MDA, and ROS levels in HepG2 cells, suggesting an increased lipid accumulation and oxidative stress. Similar to the in vitro results, mice receiving PBDE-209 and high fat showed elevated levels of sterol regulatory element-binding protein 1 (SREBP-1), stearoyl-CoA desaturase 1, and fatty acid synthase, thus promoting lipid deposition and NAFLD progression [19].

### 2.6. Others

Aside from those mentioned above, other effects and mechanisms of PBDEs on the liver are also worth mentioning. Crump et al. used an in vitro experiment to study the effects of PBDEs on cultured hepatocytes derived from embryonic chickens. They have found PBDE-71 diminished transthyretin (*TTR*), thyroid hormone–responsive spot 14-α (*THRSP14-α*), and liver fatty acid–binding protein (*FABP*) genes expression [80]. PBDE-71 can also induce hypomethylation at the T-Box Transcription Factor 3 (*Tbx3*) locus. As a transcription factor important in liver tumorigenesis, *Tbx3* hypomethylation in mouse liver cells indicated that PBDE-71 may engage in liver carcinoma development [81]. To gain knowledge about the toxicological mechanisms of PBDEs, primary Atlantic salmon hepatocytes were exposed to these congeners alone or in combination (PBDE-47, PBDE-153 and PBDE-154). Levels of endoplasmic reticulum-responsive genes vitellogenin (*VTG*) and zona pellucida 3 (*ZP3*) become elevated [82]. Early life exposure to PBDE-99 can induce hepatic inflammation and increase acetate and succinate levels [83]. To elucidate the PBDEs-gut microbiome interactions in modulating hepatic long noncoding RNAs (lncRNAs) and protein-coding genes (PCGs), conventional and germ-free mice were orally dosed with PBDE-47 or PBDE-99. LncRNAs increased the translational efficiency of PCGs, and this process might be a compensatory mechanism in response to PBDEs exposure. Pathway analysis of PCGs paired with lncRNAs displayed that PBDE-47 regulated nucleic acid, retinol metabolism and circadian rhythm, whereas PBDE-99 regulated fatty acid metabolism in conventional mice. Likewise, in germ-free mice, glutathione conjugation and transcriptional regulation were regulated by PBDE-47. In addition, the xenobiotic-metabolizing *CYP3A* genes and the fatty acid-metabolizing *CYP4* genes were modulated by PBDE-99 [84]. In Sueyoshi et al.’s study, human primary hepatocytes exposed to PBDE-47 exhibited upregulated CYP2B6 expression at both gene and protein levels. Because *CYP2B6* is a constitutive androstane receptor (CAR) target gene, the changed expression pattern suggested a cause-and-effect relationship between PBDE-47 and CAR pathway [85]. It has been reported PBDEs modulated several processes linked to pregnane X receptor (PXR) and CAR (i.e., protein ubiqutination, PPARα-RXRα activation) [86]. A further study exploring potential underlying mechanisms revealed that PBDE-209 could incur liver morphological alteration, cause oxidative stress, and subsequently reduce *PXR*, *CAR*, and CYP3A expression [87]. The effects and mechanisms of liver toxicity induced by PBDEs are shown in Table 1.

## 3. Kidney Toxicity

### 3.1. Oxidative Damage and Apoptosis

To illuminate the effects of PBDEs on the kidney, adult male rats received a gavage dose of 1.2 mg/kg PBDE-99 for the study duration of 45 days. Decreased CAT activity and increased GSSG/GSH ratio were detected after PBDE-99 exposure [48]. Human embryonic kidney cells (HEK293) were exposed to PBDE-47, and a range of bioassays were performed. For instance, PBDE-47 could change Bcl-2 family-encoding gene expression, including Bcl-2-associated death promoter (*Bad*), Harakiri (*Hrk*) and *Bcl-2*. Besides, energy metabolism disturbances characterized by altered ethanol, GSH, creatine, aspartate, uridine diphosphate glucose (UDP)-glucose and NAD^+^ were observed in PBDE-47 administration [88]. Ctenopharyngodon idellus kidney (CIK) cells treated with 100 μM PBDE-47 showed a drop in antioxidant enzymes, such as CAT, SOD, GPx, and total antioxidant capacity (T-AOC). A significant elevation in Bcl-2-associated X protein (Bax), Cytochrome c, and Caspase-3 was observed in PBDE-47 exposure compared to the normal group [89]. Consistently, a pharmacological study has shown troxerutin prevented cytochrome c release, Caspase activation, and poly ADP ribose polymerase (PARP) cleavage, raised antioxidative enzymes and Nrf2 activities, thus relieving the toxic effects of PBDE-47 on kidney [90]. Analogously, male broilers were exposed to PBDE-209 for 42 days. Swelling and granular degeneration of the renal tubular epithelium and atrophy and necrosis of glomeruli were observed. Additionally, oxidative stress indicators (MDA, GPx, GSH, SOD) were changed in the kidney [91]. Furthermore, PBDE-209 could upregulate apoptosis-related protein expression, including Bax/Bcl-2 ratio, p-extracellular signal-regulated kinase 1/2 (ERK1/2), p-JNK1/2, Bax, Cytochrome c and Caspase-3 [91]. It was reported that PBDE-209 does not affect kidney weight, while PBDE-209 supplement showed greater GSH and thiobarbituric acid reactive substances (TBARS) and reduced total -SH groups, with consequent exacerbation of nephrotoxicity [92]. The mechanisms are shown in Figure 2 and Figure 3.

### 3.2. Combined Exposures

The combined exposure of PBDE-47 and cadmium (Cd) displayed cell rounding and swelling, eventually resulting in renal tubular epithelial cell damage. Using human kidney cells (HKC), Zhang et al. reported that intracellular LDH release, nucleotide-binding oligomerization domain-like receptor protein 3 containing pyrin domain (NLRP3), cleaved Caspase-1 and cleaved gasdermin D (GSDMD) were increased by co-exposure. Further, it has been found that co-exposure to PBDE-47 and Cd could give rise to mitochondrial dysfunction NLRP3 inflammasome and GSDMD-dependent pyroptosis. Interestingly, NAC, a ROS scavenger, could mitigate the percentage of apoptotic and necrotic cells inflicted by PBDE-47 and Cd [93].

### 3.3. Others

PBDE-47 can inhibit mitochondrial fusion and fission, causing MMP decreases, ROS overgeneration, ATP depletion, and cellular disintegration in porcine kidney-15 (PK15) cell [94]. Deeper cells investigation revealed that underlying AMPK-Sirtuin 1 (Sirt1)-Peroxisome proliferator-activated receptor γ coactivator 1-α (PGC-1α) signaling pathway that might be driving the toxic changes in CIK cells subjected to PBDE-47 [89]. By using the CIK cell line, Li et al. have found PBDE-47 can enhance cytoplasmic Ca^2+^ concentration, reduce *miR-140-5p* miRNA level, increase Toll-like Receptor 4 (TLR4) and nuclear factor-κB (NF-κB) mediated inflammatory factors release. Intriguingly, melatonin could protect against PBDE-47-triggered necroptosis via targeting miR-140-5p/TLR4/NF-κB pathway [95]. Similarly, another study showed that PBDE-47-treated mice had elevated ROS, NF-κB, urine albumin-to-creatinine ratio (ACR) and NLRP3 inflammasome level, while troxerutin effectively improved kidney injury elicited by PBDE-47 through inhibiting C-X-C chemokine ligand 12 receptor 4 (CXCR4)-TXNIP-NLRP3 inflammasome [96]. The effects and mechanisms of kidney toxicity induced by PBDEs are shown in Table 2.

## 4. Gut Toxicity

### 4.1. Oxidative Damage and Apoptosis

It was reported that the metabolic activities of bacteria in the guts were impacted by PBDE-71, including functions related to energy metabolism, virulence, respiration, cell division, cell signaling, and stress response. For instance, the disruption of epithelial barrier integrity, inflammatory response and anti-oxidant capacity were observed in male intestines after PBDE-71 exposure [97]. Li et al. chose the carcinoma colon-2 (Caco-2) cells model to study the toxicity mechanism of PBDE-209. The mRNA expression of the antioxidative defense factor, *Nrf2*, was suppressed by PBDE-209. Besides, Caco-2 cells exposed to PBDE-209 exhibited a rise in Fas cell surface death receptor (*FAS*) and *CYP1A1* mRNA expression levels [98]. The mechanisms are shown in Figure 2 and Figure 3.

### 4.2. Intestinal Microbiome Disturbance

Maternal exposed to the PBDE-47 exhibited a distinctive profile in the microbiome of the gut, compared with the control dam, as shown by a decrease in genera *AF12* and *Oscillospira* and an increase in the *Actinobacteria* phylum and genera *Blautia*, *Gemella* and *Phascolarctobacterium*. Serum metabolites strongly associated with the altered gut microbiota in response to PBDE-47 are likely involved in amino acid, lipid, carbohydrate, and energy metabolism [99]. The gut microbiome plays a crucial role in toxicological responses. The intestinal microbiome was required for PBDEs to reduce 3-indolepropionic acid. A tryptophan metabolite has been demonstrated to have protective properties against inflammation and diabetes [100]. PBDE-47 continuously increased the fecal and liver bile acids in the 12α hydroxylation pathway, corresponding to an up-regulation with the hepatic bile acid-synthetic enzyme *CYP7A1* and reduced farnesoid X receptor (FXR) signaling [101]. Exposure of ICR mice to PBDE-47 in-utero and during lactation in its early life may significantly cause a drop in microbial diversity and compositional alterations, and when combined with a HFD, may further exacerbate the progression of obesity and other metabolic illnesses [102,103]. Through the action of the gut microbiome, primary bile acids are converted into more lipophilic secondary bile acids that may be taken up by the host and interact with certain receptors. Both PBDE-47 and PBDE-99 decreased the proteins of sodium taurocholate cotransporting polypeptide (Ntcp) and organic anion transporting polypeptide 1b2 (Oatp1b2) in a gut microbiota-dependent manner [104]. Neonatal contact with PBDE-99 caused a lasting rise in *Akkermansia muciniphila* throughout the intestine, along with augmented hepatic levels of acetate and succinate, the expected byproducts of *A. muciniphila* [83]. The mechanisms are shown in Figure 6. The effects and mechanisms of gut toxicity induced by PBDEs are shown in Table 3.

## 5. Thyroid Toxicity

### 5.1. Hormonal Interferences

Huang et al. have demonstrated that even low concentrations of PBDEs could potentially affect THs in the general population [105]. Research has sought to fill the void by establishing a human PXR-overexpressing HepG2 cell model and a dual luciferase reporter assay system to examine the involvement of hPXR in the modifications of thyroid receptor (TR) expression caused by PBDE-47 in HepG2 cells. TR isoforms (TRα1 and TRβ1) were both observed to decrease in both mRNA and protein expression when the concentration of PBDE-47 was increased in HepG2-pCI-hPXR-neo cells, which may provide more evidence for the toxicological mechanisms of the disruption of the TH in the presence of PBDE-47 [106]. Consistently, Macaulay et al. have demonstrated that PBDE-47 negatively affected the early development of the zebrafish by reducing the TR [107]. Intriguingly, in larvae, PBDEs (PBDE-47 and PBDE-209) significantly stimulated several genes, which included *TRα* and *TRβ*, thyroglobulin (*TG*), thyroid peroxidase, *TTR*, corticotrophin-releasing hormone (*CRH*), sodium iodide symporter (*NIS*), thyronine deiodinase (*Dio1* and *Dio2*), uridinediphosphate-glucuronosyl-transferase (*UGT1ab*) and thyroid stimulating hormone (*TSH*) [108,109]. Lower plasma T4 and liver vitamin A concentrations were linked to PBDE-71 exposure [110]. Concentrations of total and free total thyroxine (FT4) were significantly decreased by PBDE-71 in a dose-dependent manner [111]. A positive link between TSH and almost all PBDE congeners was identified, while an inverse relationship between PBDE-154 and free triiodothyronine (FT3) and FT4 was found [112]. Conversely, lower TSH levels have been linked to exposure to PBDEs in pregnant women [2,113]. Interestingly, there was a significant positive relationship between serum PBDE-209 levels and total thyroxine (TT4), as well as a marginal positive relationship with total triiodothyronine (TT3), in all occupational workers after accounting for gender, age, body mass index (BMI), and duration of occupational exposure [114]. Exposure to PBDE-209 changed the thyroid gland’s structure [115]. Hydroxylated PBDEs (OH-PBDEs) have a close structural affinity to TH, and have been demonstrated to interact antagonistically with human TTR, a T4 transport protein [116]. Consistently, changes in plasma FT4 levels in rainbow trout plasma, potentially caused by the metabolic activity of PBDE 209, might be due to the competition of OH-PBDEs for binding sites on TTR [117]. Analogously, Ibhazehiebo et al. have reported that PBDEs diminished TR-mediated gene expression by partially separating TR from TH response element (TRE) through the DNA binding domain (DBD) [118]. Fish exposed to PBDE-209 exhibited a drop in circulating TT4 and TT3 compared to controls [119]. CAR/PXR pathways may be the underlying cause of the decrease in TT4 following PBDE-47 exposure, which is evident in the elevated UGT activity and inducibility of genes in the CAR/PXR pathway, namely *CYP2B10*, *Ugt1a1*, *Ugt1a7*, *Ugt2b5* and multidrug resistance protein-associated protein (*Mrp3*) [120,121]. The mechanisms are shown in Figure 7.

### 5.2. Oxidative Damage and Apoptosis

PBDE-47 augmented apoptosis in thyroid tissue, as revealed by Caspase-3 activation, PARP cleavage and DNA fragmentation. Additionally, the increased glucose-regulated protein 78 (GRP78), activating transcription factor 4 (ATF4), Caspase-12, C/EBP homologous protein (CHOP) and sequestome 1 (p62) accumulation were observed. These results indicate that excessive ER stress, defective autophagy and the resultant apoptosis are thought to be involved in maternal thyroid harm after perigestational PBDE-47 exposure [122,123,124]. Besides, further studies showed that oxidative damage and hypothalamic-pituitary-thyroid (HPT) axis-related gene alterations may be the underlying mechanisms involved in the toxicity of PBDEs (PBDE-47, PBDE-71, PBDE-99, PBDE-209) [125,126,127,128,129,130,131,132,133,134]. The mechanisms are shown in Figure 2 and Figure 3.

### 5.3. Indirect Exposures

There was a correlation between elevated levels of maternal serum PBDEs 28 and 47 and increased maternal serum concentrations of T4 and T3 during the early second trimester of pregnancy [135]. Conversely, several studies revealed a negative association between hormones (T4 and T3) and PBDEs [136,137,138,139,140]. Positive associations between PBDEs and THs were found in the high-exposed population, while negative associations were reported in the low-exposed populations [141,142]. Discrepancies in the direction of correlations have been noted, and the potential explanations could include the low-dose and nonmonotonic effects of endocrine-disrupting chemicals [143]. The treatment of mothers with PBDE-71 affects the first filial generation (F1) females, as shown by a decrease in body weight and elevated serum T3 and T4 levels. In addition, thyroid gland weight and ovarian osteopontin mRNA were increased at five months of age [144]. Among infants delivered naturally and unassisted via vaginal delivery, prenatal polychlorinated biphenyls (PCBs) and PBDE exposures were associated with lower TT4 and FT4 levels [145].

### 5.4. Combined Exposures

Results showed that the co-exposition of polystyrene nanoplastics (PS-NPs) and the PBDE-47 aggravated the deformity in pericardial edema, yolk sac edema and curvature of the tail in the larvae of zebrafish. Interestingly, an investigation of the HPT axis-related genes showed that the expressions of *TSHβ*, *TG*, *Dio2* and *TRβ* were increased more prominently when both PBDE-47 and NPs were present, compared to PBDE-47 single exposure [146].

### 5.5. Others

Expression of Dio3 in placental cells is disrupted by low-dose PBDE-209, resulting in interference of TH. Modifications in the miRNA expression pattern of the miR379/656 cluster in the delta-like homolog 1 (Dlk1)-Dio3 imprinting domain, particularly of miR409-3p and miR668-3p, and/or changes in the DNA methylation of the cells, specifically the intergenic-differentially methylated regions (IG-DMR) and maternally expressed gene 3-DMR (MEG3-DMR) in the Dlk1-Dio3 imprinting region, may be responsible for the disturbance in Dio3 expression brought about by PBDE-209 [147]. The effects and mechanisms of thyroid toxicity induced by PBDEs are shown in Table 4.

## 6. Embryotoxicity

### 6.1. Oxidative Damage and Apoptosis

In vivo and in vitro models have shown that PBDE-47 could activate the MAPK signaling pathway, thus changing impaired placental physiological function [148]. Embryonic development abnormalities in zebrafish exposed to PBDE-47 could be improved by ROS scavenging and JNK inhibition. Therefore, deficiencies in mitochondrial biogenesis and mitochondrial dynamics, which may lead to ROS/JNK activation, could be the reason for PBDE-47-induced developmental toxicity [149,150]. Meanwhile, supplementation with the antioxidant NAC could partly reverse the ROS increase and octamer-binding transcription factor 4 (OCT4) downregulation caused by PBDE-209 exposure [151]. It was reported that PBDE exposure caused a decrease in the expression of pluripotent genes such as *OCT4*, SRY-box transcription factor 2 (*SOX2*) and Nanog homeobox (*NANOG*) and prompted apoptosis in embryonic stem cells (ESCs) [151,152]. The mechanisms are shown in Figure 2 and Figure 3.

### 6.2. Combined Exposures

Exposure to multiple chemicals is a common occurrence in the environment. For example, embryos temporarily postpone hatching when encountering PBDE-209 and nSiO_2_ at 60 h post fertilization. PBDE-209-nano-SiO_2_ co-exposure showed a decreased heartbeat and increased arrhythmia in zebrafish larvae compared to individual PBDE-209 treatments [153].

### 6.3. Others

Embryonic developmental abnormalities of SD rats (e.g., soft tissue syndactyly or malposition of the distal phalanges and decreased ossification of the sixth sternebra), zebrafish (e.g., embryo yolk sac, pericardial edema, spine deformation, neurobehavioral abnormalities and blood vessels damage) and kestrels (e.g., shorter humerus length and reduced total thyroid weight) arise as a result of being exposed to PBDEs [154,155,156,157,158]. The Dio3 is of great significance in maintaining the fetal thyroid equilibrium. PBDE-47 correlated with an increase in placental *Dio3* methylation, whereas this effect was only observed in female infants [159]. Epidemiological studies have indicated that PBDE-47 can bring about adverse pregnancy results. PBDE-47-treated mice displayed decreased vascular endothelial growth factor-A (VEGF-A) protein expression, indicating that PBDE-47 may disrupt placental angiogenesis [160]. Transcriptomic analysis of the PBDE-47 effect suggested concentration-dependent changes in the expression of genes, including stress pathways such as inflammation and metabolism of lipids/cholesterol, as well a process underlying the fate of trophocytes, such as differentiation, migration, and morphology of the vascular system [161]. Mediation analyses revealed that PBDEs exposure could potentially impact fetal growth through insulin-like growth factor 2 (IGF2) methylation [162]. Chi et al.’s multivariate statistical analysis revealed a marked change in the metabolic profile resulting from PBDE-209 embryotoxicity in maternal serum. Administration of PBDE-209 at a dosage of 2500 mg/kg caused considerable disturbances to thyroid hormone metabolism, the tricarboxylic acid cycle (TCA) cycle, and lipid metabolism in maternal mice, leading to substantial inhibition of fetal growth and development [163]. Consistently, Du et al.’s study demonstrated that prenatal PBDE-209 induces upregulation of endothelin-1 (ET-1) and inducible nitric oxide synthase (iNOS) and downregulation of endothelial nitric oxide synthase (eNOS) in the placenta, which in turn is associated with reduced birth weight of the newborns [164]. Gestational exposure to PBDE-209 can reduce placental weight, impede placental vascular growth, and cause cell death. Mechanically, gestational exposure to PBDE-209 augmented the expression of GRP78 and activated pancreatic endoplasmic reticulum kinase (PERK) signaling [165]. The exposure to PBDE-47, PBDE-99 and PBDE-209 at realistic concentrations led to lethal and sublethal changes and impaired gene expression involved in TH homeostasis, including *TSHβ*, *TTR*, thyroxine-binding globulin (*Tbg*) and *Dio1*, leading to an abnormal development of zebrafish embryos [166]. The effects and mechanisms of embryotoxicity induced by PBDEs are shown in Table 5.

## 7. Reproductive Toxicity

### 7.1. Oxidative Damage and Apoptosis

Oxidative stress and reduced testosterone levels induced by PBDE-209 can cause DNA damage and activate the ataxia-telangiectasia-mutated (ATM)/checkpoint kinases 2 (Chk2), ataxia telangiectasia and Rad3-related (ATR)/checkpoint kinases 1 (Chk1) and DNA-dependent protein kinase catalytic subunit (DNA-PKcs)/X-ray repair cross-complementing protein 4 (XRCC4)/DNA ligase IV pathways [167,168]. Furthermore, PBDE-209 may damage the mitochondrial function by reducing the length of the telomeres, decreasing the activity of telomerase, and activating PPARγ/RXRα/sterol regulatory element-binding protein cleavage-activating protein (SCAP)/SREBP-1 pathway, resulting in cell apoptosis [169,170]. Following maternal exposure to PBDE-209 during lactation, prepubertal mice offspring exhibited impaired germ cell proliferation, affected steroidogenesis and increased germ cell apoptosis, alongside modifications in the expression of various cell survival and apoptotic markers and a reduction in the expression of gap junction connexin 43 (cx43) and cyclin-dependent kinase inhibitor 1B (p27Kip1) [171,172,173,174,175]. Similarly, a substantial drop in the net reproductive rate and intrinsic increase rate transpired when the concentration of PBDE-47 was high. PBDE-47 had a strongly detrimental effect on the ultrastructure of the ovary. Further studies showed that PBDE-47’s reproductive toxicity was attributed to the ROS-mediated pathway [176,177]. PBDE-47 induced mitochondrial disruption (e.g., aberrant distribution and diminished MMP), which can induce apoptosis of early leptotene spermatocytes and affect the maturation of oocytes [178,179]. GC2 (immortalized mouse spermatocyte) exposed to PBDE-47 have reduced cell viability, and condensation of nuclear and vacuolated mitochondria. Meanwhile, the deletion of ATP synthase subunit β (Atp5b) or ubiquinol-cytochrome-c reductase complex core protein 1 (Uqcrc1) led to a decrease in MMP and triggered apoptosis in GC2. This may be part of the underlying mechanism of the association between PBDE-47 and spermatocytes [180]. PBDE-99 could induce Leydig cell apoptosis via increasing ROS, triggering the ERK1/2 pathway, and inhibiting the ubiquitination degradation pathway [181]. PBDEs (PBDE-47, PBDE-99 and PBDE-100) are capable of initiating apoptosis, both through the extrinsic and intrinsic pathways, upon extended exposure periods; this can cause early malfunction of the corpus luteum [182]. PBDE-3 decreased serum testosterone levels and Leydig cell size by decreasing extracellular signal-regulated kinase 1/2 (ERK1/2), AKT, and AMPK phosphorylation and elevating ROS production [183,184]. Caenorhabditis elegans (worm) that underwent PBDEs (PBDE-3 and PBDE-47) treatment showed reduced life spans, impeded fecundity, and delayed egg-laying. Most interestingly, mutants of C. elegans p53-like protein (*cep-1*), DNA damage checkpoint proteins (*hus-1*) and mitogen-activated protein kinase (*mek-1* and *sek-1*) rescinded the germ cell apoptosis induced by PBDEs [185,186]. The mechanisms are shown in Figure 2 and Figure 3.

### 7.2. Epigenetic Inheritance

Moreover, the effects on reproductive health may even be transgenerational. For instance, PBDE-47 treatment displayed that protamine and transition protein genes were, on average, reduced by four-fold in the testes, suggesting that histone-protamine exchange could be disrupted during spermatogenesis, triggering an aberrant sperm epigenome [187]. According to Suvorov et al., sperm samples collected on PND 65 and PND 120 yielded 21 DMRs and 9 DMRs, respectively [188]. Following exposure to PBDE-209, there was a decrease in anogenital distance and the percentage of sperm with normal morphology. Further exploration revealed that, in contrast to the control group, the PBDE-209 group had 215 genes exhibiting hyper-methylation and 83 genes displaying hypo-methylation [189]. The GC-2spd mouse spermatocyte line was utilized to analyze the poisonous effects of PBDE-209 on methylation and spermatogenesis. The results indicated that PBDE-209-induced spermatogenesis damage was due to its disturbance of SET domain-containing protein 8 (SETD8)/Histone H4K20 monomethylation (H4K20me1)-linked histone methylation, inhibition of meiosis initiation and cell cycle progression, which in turn caused long-term male reproductive toxicity [190,191].

### 7.3. Mitochondrial Damage

PBDEs showed a negative relationship with semen mobility and sperm quality [192,193,194,195]. For example, PBDE-47 likely interferes with the ER-Nrf1-mitochondrial transcription factor A (Tfam)-mitochondria pathway, thereby reducing mitochondrial function, impairing spermatogenesis, and damaging germ cells [196]. Shan et al.’s findings unveiled that PBDE-47 impaired mitochondrial function and cholesterol transport, ultimately leading to a reduction in progesterone synthesis [197]. After administering a small amount of PBDE-99, Talsness et al. noticed adverse ultrastructural changes in the mitochondria of the F1 female offspring’s ovary [198]. The mechanisms are shown in Figure 5.

### 7.4. Combined Exposures

In male rats treated with PBDE 47, HFD exacerbated the damage to the seminiferous epithelia and decreased testosterone levels, which decreased the number of spermatozoa. Further mechanistic exploration revealed HFD triggered PBDE47-induced dosage-sensitive sex reversal adrenal hypoplasia congenital critical region on X chromosome gene 1 (DAX-1) expression and lowered steroidogenic acute regulatory protein (StAR) and 3β-hydroxysteroid dehydrogenase (3β-HSD) levels in rat testicular interstitials [199].

### 7.5. Others

The effect of PBDE-209 on blood-testis barrier ultrastructure was destructive, with the destruction of tight junctions, ectoplasmic specialization structures with broken tight junctions, and a lack of actin microfilaments [200]. Moreover, the elevation of estrogen receptor signaling caused by PBDE-209 leads to disruption of the blood testis barrier in male mice of the F1 generation [201]. PBDEs disrupted gonadal development and had a reduced fecundity [202,203,204,205,206]. In adult mice, PBDE-209 treatment caused a diminished sperm quality and arrested meiotic prophase I at the early-pachytene stage during spermatogenesis [207,208,209]. Postnatal exposure to PBDE-209 at a low dose from day one to day five results in lower testosterone and androgen receptor (Ar) and TRα transcripts in Sertoli cells, along with an imbalance in the TRα splicing variants ratio, causing a decreased testicular size and hindered spermatogenesis [210]. Analysis of 42 differentially expressed proteins in the testis revealed that downregulating histone variants and parvalbumins associated with PBDE-47 may impede spermatogenesis and lead to infertility in fishes. The increase in VTGs and apolipoprotein A–I suggested that PBDE-47 acts like a mimicker of estrogen and causes reproductive dysfunction [211,212]. At higher concentrations, PBDE-47 caused prolonged hyperactivation of autophagy, which ultimately caused ovary damage [213]. In addition, fish exposed to PBDE-47 during early life stages had reduced clutch size and lower fecundity than the control group [214]. Interestingly, PBDE-71 significantly increased malformation and the percentage of males in the F1 generation and reduced frequencies of male courtship behaviors [215,216]. Arowolo et al. indicated that PBDEs and their metabolites, when present at environmental levels, may impact male reproductive health through AR antagonism, testosterone signaling, cAMP production, mechanistic target of rapamycin complex one (mTORC1) signaling and TH transport [217]. In addition, it appears that PBDE-47 has the potential to enhance the sensitivity of adult Leydig cells to cAMP when synthesizing androgen [218]. The effects and mechanisms of reproductive toxicity induced by PBDEs are shown in Table 6.

## 8. Neurotoxicity

### 8.1. Apoptosis

A concentration-dependent increase in the protein expression of Fas and Fas-associated death domain (FADD), as well as activation of Caspases (Caspase-8 and Caspase-3), was detected, implying involvement of the death-receptor pathway in the PBDE-209-induced Neuro-2a cell apoptosis [219]. By increasingthe expression of phosphodiesterases (*PDEs*) that modify intracellular cyclic guanosine monophosphate (cGMP) levels and reducing the Bcl-2/Bax ratio, apoptosis induction was induced by PBDE-209 [220]. The autophagy proteins, such as microtubule-associated protein-I light chain 3 (LC3)-Ⅱ and Beclin-1, and apoptosis proteins, including Bcl-2, Caspase-3 and PARP, were changed after PBDE-209 administration. This modulation could reduce the learning and memory capabilities of the offspring [221]. Adult rats exposed to PBDE-153 exhibited impaired learning ability, reduced spontaneous activity and neuron apoptosis [222]. Costa et al. have found that antagonists of glutamate ionotropic receptors reduced the toxicity of PBDE-47 in mouse cerebellar neurons, suggesting PBDE-47 may heighten extracellular glutamate, which then stimulates ionotropic glutamate receptors and brings about increased calcium levels, oxidative stress, and finally, cell death [223,224]. PBDE-47 has increased Caspase-3, Caspase-12 and cytochrome c levels in the rats’ hippocampus [225]. Cytotoxic evaluation has indicated that PBDE-99 exhibits cytotoxicity against rat cerebellar granule neurons (CGNs). Besides, a decrease in the expression of brain-derived neurotrophic factor (BDNF) and Bcl-2 was also detected after PBDE-99 treatment [226]. The mechanisms are shown in Figure 3.

### 8.2. Disease Induction

Consistent with the cytotoxicity reported above, perinatal exposure to PBDE-99 through gestation and ingestion of maternal breast milk may lead to learning difficulties, BDNF downregulation and free radicals’ production in the offspring of rats [227,228]. Similarly, it was reported that PBDE-47 could reduce BDNF production and increase the risk of post-partum depression [229]. Besides, PBDEs (PBDE-209, PBDE-206 and PBDE-203) were shown to reduce BDNF concentration and increase calcium/calmodulin-dependent protein kinase II (CaMKII) levels in mice hippocampus [230,231]. There is a significant correlation between PBDE-47 and PBDE-99 exposures and depression symptoms among the pregnant cohort [232]. Perinatal exposure to PBDE-47 has been found to decrease the length of the dendrites, the complexity of the branching patterns, and the density of the spines in the prefrontal cortex of offspring. These effects may contribute to autistic behavior [233]. Wang et al. have found that PBDE-71 leads to a significant decrease in serotonin levels and levels of tyrosine hydroxylase and dopamine transporter proteins [234,235]. 6-OH-PBDE-47 is a highly metabolized form of PBDE-47 in vivo. 6-OH-PBDE-47 administration could induce motor defect by impairing the dopaminergic system and promote α-synuclein aggregation by inhibiting ubiquitination and autophagy, indicating that the presence of 6-OH-PBDE-47 in the brain could pose a risk for developing Parkinson’s disease (PD) [236].

### 8.3. Intestinal Microbiome Disturbance

PBDE-47 exposure during gestational and lactational periods displayed hyperactivity and anxiety-like behavior. Furthermore, 16S rRNA sequencing of fecal samples revealed a distinctive community composition of gut microbes after exposure to PBDE-47, which manifestes as a decrease in genera *Ruminococcaceae* and *Moraxella*, and an increase in genera *Escherichia-Shigella*, *Pseudomonas* and *Peptococcus*. Qiu et al. have discovered that the changes in the intestinal flora are involved in the alterations in serum metabolite levels, and both are correlated with locomotion hyperactivity and anxiety [237]. The mechanisms are shown in Figure 6.

### 8.4. Combined Exposures

When exposed together, PS-NPs and PBDE-47 coalesced into bigger particles. Neurodevelopmental toxicity (e.g., accelerated voluntary movements) in zebrafish larvae was heightened with concurrent exposure to PS-NPs and PBDE-47. Besides, the expression of the acetylcholinesterase (*ache*) and the cholinergic receptor nicotinic 7 α subunit (*chrn7*α) genes, which are associated with the development of neurocentral cells, was significantly decreased, mainly in the co-exposure group [79]. Analogously, the chemical mixtures (PBDE-47, 6-OH-PBDE-47 and 6-MeO-PBDE-47) caused a decrease in AChE activity, implying the potential neurological responses of such treatment [238]. Generally, the brain obtains free fatty acids from the systemic circulation and further alters them into structural and signaling lipids to guarantee proper neurotransmission [239]. A mixture of PCBs and PBDEs (MIX) caused neurobehavioral defects, and further studies observed impaired mitochondrial function and lipid metabolism regulation [239,240]. Pregnant C57BL/6J female mice were exposed to PBDE-209/Pb mixture. The male offspring have increased pro-inflammation cytokines, such as tumor necrosis factor α (TNFα), interferon γ (IFNγ), interleukin 4 (IL-4), interleukin 6 (IL-6), interleukin 10 (IL-10) and interleukin 17 A (IL-17 A) in the serum. Moreover, the male offspring displayed decreased neuronal cells in the cornu ammonis 1 (CA1) and CA3 subregions of the hippocampus and impaired learning behavior [241]. Meanwhile, ROS scavenger NAC can reduce locomotor dysfunction induced by co-exposure (PBDE-209 and Pb), suggesting ROS may be a major factor in eliciting developmental neurotoxicity [242]. Analysis of chemicals revealed that PBDE-209 was taken up and processed by zebrafish larvae, and the presence of titanium dioxide nanoparticles (nano-TiO_2_) increased the rate of PBDE-209 absorption. The joint presence of nano-TiO_2_ and PBDE-209 decreased locomotion activity and downregulation of specific genes and proteins related to the central nervous system of developing zebrafish larvae [243]. PBDE-99, in combination with methylmercury, augmented developmental neurotoxic effects, including impaired negative geotaxis reflexes and motor coordination [244].

### 8.5. Others

Chen et al. indicated common PBDE congeners might be toxic agents in neural precursors, which cause functional changes and induce transcriptome changes in pathways that regulate neurodevelopment, hormone signaling, and the response to stress in the environment [245]. PBDEs (PBDE-99 and PBDE-47) exposure in the neonatal period disrupts the normal development of the brain and causes a disturbance in spontaneous behavior [246,247,248]. In the cortex, a high level of growth-associated protein-43 (Gap 43), a neuronal growth-related protein, was observed [246,247]. PBDE-47 increased the spontaneous coil activity in the embryos of zebrafish under high-intensity light and decreased the locomotion in the larvae of zebrafish. These locomotion effects were negatively correlated with tissue PBDE-47 levels and might be related to pathways for early neurogenesis, the central nervous system and development of the axes [249,250]. PBDE-47 could interfere with neurogenic locus notch homolog protein (NOCTH), GSK3β and T3 signaling, which may affect neurogenesis [251]. Azar et al. have reported that prenatal PBDE exposure is correlated with a decrease in cognitive ability in preschool-age boys, but no such association was seen in girls at the concentrations of exposure in Canada [252]. It has been established that primary sensory neurons are susceptible to the neurotoxic effects of PBDE-209 [253]. C57BL/6J mice were given an oral dose of 20 mg/kg PBDE-209 from day 1 to 21. A drop in TH and/or glial cell activity could impede hippocampal growth, resulting in behavioral difficulties [254]. PBDE-99 was provided to CD-1 Swiss females orally daily from gestational day (GD) 6 to PND 21. On PND 60, the treated mice exhibited an altered thigmotaxis, devoting more time to the centre of the arena than the control mice [255]. PBDE-99 inhibits the differentiation of a mouse and human neural progenitor cell (NPC) lineage into a lineage of oligodendroglial based on species-specific actions [256]. Startle reactions to acoustic stimuli were intensified by PBDE-71 at PND 90, displaying the delay of sensory responsiveness [257]. In zebrafish larvae, hyperactivity was seen when PBDE-71 was present in low concentrations, whilst higher concentrations led to decreased activity during the dark period [258]. Disruption of calcium balance can be caused by PBDE-71, resulting in decreased cholinergic function and locomotor activity [259]. Perinatal exposure to PBDE-71 induced transcriptional alterations, including neurofilaments and cell adhesion molecules (i.e., N-cadherin and CAMKII, and cytokines) [260]. PBDE-47 exposure drastically affects spontaneous movements, decreases the response to touch and speed of swimming, and alters larvae’s swimming behavior due to light stimulation. The inhibition of the axonal growth of primary and secondary motor neurons was found, which may contribute to these motor deficits [261]. Hedgehog signaling, a pathway involved in the development of embryos and neurogenesis, was suppressed due to PBDE-47 [262]. PBDE-47 and PBDE-99 could cause short- and long-term behavioral damage at low exposure levels [263]. Additionally, PBDEs (PBDE-47 and PBDE-49) delayed neuronal polarization, leading to a substantial decrease in axonal outgrowth within the first few days in vitro. Reduced ryanodine receptor (RyR) activity could block these axon inhibitory effects, indicating that a potential RyR-dependent mechanism is involved in PBDEs neurotoxicity [264]. The expression of two G1-phase-related regulatory factors genes, *p53* and cyclin-dependent kinase inhibitor 1 (*p21*), was significantly increased by PBDE-47 treatment. On the other hand, reduced gene expression of the *cyclin D1* and the cyclin-dependent kinase 2 (*CDK2*) occurred after PBDE-47 exposure [265]. Besides, PBDE-47 was reported to reduce the MMP and increase the release of cytochrome c to the cytoplasm [266]. Therefore, these processes might cause the reduced Neuro-2a cell proliferation [219,265,266]. PBDE-209 exposure at different developmental stages (i.e., pregnancy, lactation, intragastric administration, after weaning and prenatal to life) could alter the synaptic plasticity in adult rats [267]. PBDE-209 significantly affected dendritic branch number, synaptic protein levels and doublecortin in neurons [268,269]. PBDE-209 exposure to pregnant and lactating mice can disrupt the serum THs of the offspring, as it alters the expression of the Dio, thus resulting in neural impairment [270]. Calcium overload plays a vital role in neuronal function. After being subjected to PBDE-209, the concentration of Ca^2+^ in the hippocampus of the offspring was increased and impaired learning and memory occurred [271]. Roberts et al. have reported that PBDEs (PBDE-99 and PBDE-47) could reduce Dio2 activity in primary astrocyte cells and H4 glioma cells, which consequently caused neurodevelopmental deficits [272]. Using the Gesell Developmental Schedules (motor, adaptive, language, and social domains), researchers have found that prenatal PBDE exposure was linked to lower developmental quotients (DQs) in young kids [273]. The effects and mechanisms of neurotoxicity induced by PBDEs are shown in Table 7.

## 9. Immunotoxicity

### 9.1. Oxidative Damage and Apoptosis

PBDE-47 has been shown to significantly induce the formation of neutrophil extracellular traps (NETs), a central player in innate immune responses, and the mechanism may be linked to ROS [274]. Zhou et al. have shown that PBDE-47 could diminish the phagocytic ability and the bacteriolytic activity of *R. philippinarum* and *blue mussel mytilus edulis*. Further exploration revealed these changes may be related to the ROS imbalance, the MAPKs pathways, and the lysosomal membrane damage [275,276]. PBDE-47 and PBDE-209 have been reported to enhance ROS production and decrease GSH levels [277]. Exposure of harbour seal granulocytes to PBDEs (PBDE-47, PBDE-99 and PBDE-153) leads to oxidative stress by reducing thiols levels and increasing ROS production [278]. The mechanisms are shown in Figure 2 and Figure 3.

### 9.2. Inflammatory Response

PBDE-47 can modulate the expression of an array of intracellular miRNAs, which are primarily involved in the regulation of the innate immunity response [279]. Persistent exposure to PBDE-47 can impair innate immunity in mammary tissue [280]. PBDE-47 can disrupt the secretion of proinflammatory cytokines (IL-6 and TNF-α) and interfere with basophil activation [281,282,283,284]. Peripheral blood mononuclear cells (PBMC) from subjects who had autism spectrum disorders (ASD) showed a higher response to lipopolysaccharide (LPS) when pretreated with PBDE-47 compared with the control group [285]. Consistently, resistance to the pathogen was compromised in minnow and rainbow trout after exposure to PBDE-47 [286,287]. Mice exposed to PBDE-209 were deprived of proliferative effects and the production of cytokines (IL-2, IFN-γ and TNF-α) in clusters of differentiation 4 (CD4) T cells and CD8 T cells [288,289]. PBDE-209 could induce immunotoxicity, which is characterized by atrophying immune organs, altering humoral and cellular immunity and gene expression [290,291]. Broiler chicks were supplied with PBDE-209, and the subsequent histopathologic examinations showed damaged and necrotic lymphocytes in the spleen and bursa and losses of lymphoid cells in the thymic gland. Interestingly, KEGG database analysis revealed that the cytokine-cytokine receptor interaction signal pathway was most significantly enriched [292]. Mated female C57BL/6J mice were orally administered with PBDEs, and a considerably reduced number of splenocytes and thymocytes were observed in offspring, suggesting PBDEs transferred from the dam affect the offspring’s immune system [293]. Exposure to PBDEs (PBDE-47, PBDE-100 and PBDE-209) led to increased inflammatory cytokine expression, prostaglandin E2 (PGE2) levels, cAMP concentration and cyclooxygenase 2 (*COX-2*) gene expression, which illuminated that PBDEs-induced immune response might be attributed to PGE2-prostaglandin E receptors (EPs)-cAMP-cytokines signaling [294].

### 9.3. Combined Exposures

Mixtures of compounds, including carbamazepine (CBZ), CdCl_2_, and PBDE-47, have a significant impact on head-kidney (HK) leucocyte cellular activities and the humoral response in the serum or skin mucus, as well as the dysregulation of pro-inflammatory factors [295]. The effects and mechanisms of immunotoxicity induced by PBDEs are shown in Table 8.

## 10. Others

### 10.1. Diabetes Induction

An elevated prevalence of diabetes was observed in individuals who had been environmentally exposed to PBDE-47 [296]. The prevalence of gestational diabetes (GMD) is increasing rapidly in the world. It has been reported that PBDEs (PBDE-153, PBDE-154, etc.) could disturb maternal glucose homeostasis, augmenting the risk of gestational diabetes mellitus [297]. Interestingly, PBDE-209 has been shown to elevate blood glucose concentration and reduce insulin receptor (*InsR*) mRNA in HFD-fed mice [298].

### 10.2. Heart Toxicity

For 28 days, rats were supplied with corn oil containing PBDE-209 (5, 50, 500 mg/kg/day) orally. It was determined that PBDE-209 caused damage to the morphology and ultrastructure of the heart and abdominal aorta, increases in serum creatine kinase and LDH values, and changes in antioxidant enzyme activity [299]. PBDE-209 could stimulate the generation of ROS. Subsequently, ROS activated ER stress-related inositol-requiring enzyme 1α (IRE1α)/AKT/mTOR signaling pathway and elicited vascular endothelial cells apoptosis [300].

### 10.3. Eye Toxicity

To investigate the mechanism of PBDEs’ effects on visual behavior, zebrafish embryos were subjected to PBDE-71. The histological analysis uncovered a diminished area of the inner plexiform layer, an enlarged inner nuclear layer, and a reduced density of ganglion cells in the retina of the PBDE-71 group. Behavioral tests unveiled that zebrafish larvae displayed hyperactive responses (i.e., enhanced saccadic eye movements and phototactic response) to PBDE-71. Further biochemical analysis demonstrated that the patterns of mRNA and protein expression of opsin differed between normal and PBDE-71 groups [301]. It was reported that the thickness and area of the neural retina of human embryonic stem cell-derived retinal organoids (hESC-ROs) were diminished dose- and time-dependent due to exposure to PBDE-47. PBDE-47 exposure was associated with decreased cell proliferation, augmented cell apoptosis, aberrant differentiation and changed eye morphogenesis [302,303]. Moreover, metabolomics research of hESC-ROs demonstrated pronounced fluctuations in the metabolism of purine and glutathione after five weeks of PBDE-47 exposure [302]. It was reported that PBDE-71 increased the retinal and retinyl ester content in the eye of zebrafish [304]. Consistently, there has been a notable rise in the transcription of retinal dehydrogenase (*raldh2*) and a marked reduction in the transcription of retinol dehydrogenase (*rdh1*), cellular retinoic acid binding protein (*crabp1α* and *crabp2α*), retinol-binding protein (*rbp1α*) and retinoic acid receptor subunit (*raraa*) in zebrafish larvae, suggesting PBDE-71 exposure caused a disturbance of retinoid signaling [305].

### 10.4. Lung Toxicity

Inhaling PBDEs found in the air or dust can have detrimental effects on human health, particularly the respiratory system. For example, results indicated that PBDE-209 exposure can increase the leakage of LDH, inhibit the viability of cells, and increase the transcripts and protein levels of the inflammatory markers (IL-6 and IL-8) in human lung epithelial cells [306]. PBDEs inhalations may impair the health of the lungs through inflammation, oxidation stress, damage to the barrier integrity, uncontrolled production of mucous, and alterations in the physics and biochemical properties of airway fluids [307,308]. Normal human bronchial epithelial cells exposed to PBDEs (i.e., PBDE-47, PBDE-99 and PBDE-209) have been shown to cause oxidative stress, such as NADPH oxidase-4 (NOX-4) expression and ROS. A notable rise was noticed in the activation of DNA damage and repair processes (i.e., increased comet tail length and elevated histone family member X phosphorylation at ser139 site) [309]. The presence of PBDEs (PBDE-28, -47, -99, -100, -153, -154, -183, and -209) triggered membrane disruption and a release of LDH and also caused oxidative stress in cells, which was marked by a decrease in MMP and an emergence of ROS [310]. The effects and mechanisms of toxicity (diabetes, heart, eye, and lung) induced by PBDEs are shown in Table 9.

## 11. Discussion and Conclusions

As a class of typical persistent organic pollutants, PBDEs are among the most important brominated flame retardants classes, often used in plastic, textile, and electronic components and circuits, which are found in the environment through different paths and processes [311]. Food processing and packaging can lead to the contamination of food with PBDEs, thereby entering the food chain [312]. PBDEs are dispersed throughout different environmental media (water, soil, air, and dust) and pose serious threats to human health via multiple routes of exposure (ingestion, dermal, and inhalation) [313,314]. PBDEs and their metabolites have been proven to be hazardous to humans. The debromination of PBDEs occurs in organisms, resulting in a notable increase of congeners with fewer bromines [315]. PBDEs undergoing phase Ⅰ metabolism give rise to OH-PBDEs metabolites in animals, which can cause more severe biological repercussions [33]. The OH-group’s transformation into a MeO-group (i.e., MeO-PBDEs) is a phase II reaction. Nevertheless, metabolic processes do not yield significant changes in the elimination of PBDEs. Both OH- and MeO-PBDEs exhibit a relatively hydrophobic character, causing them to remain within the organism [316].

Considering the hazardous effects of PBDEs, it’s necessary to take measures to prevent their release into the environment. An effective approach to minimize e-waste containing PBDEs in landfills is to classify and disassemble electronic devices for treatment using microbial technologies that can absorb and decompose toxic substances. Besides, anti-seepage systems and the continuous monitoring of PBDEs in their leachates are needed [317]. Interestingly, mosses can be utilized to track spatial patterns and temporal trends of atmospheric concentrations or deposition of PBDEs. This may offer a low-cost, feasible approach for facilitating the timely control and sustainable management of PBDEs pollution [318]. By using nanoscale zerovalent iron (nZVI) and palladized nZVI (nZVI/Pd), it’s possible to debrominate lower-brominated PBDEs into diphenyl ether (DE), which is the fully debrominated form [319]. PBDEs mainly exist in the particulate phase of wastewater. Thus, sedimentation technology applied in wastewater treatment plants is also a vital method to remove PBDEs [320].

Various PBDE congeners and their metabolites may exert various biological effects by acting on different molecular targets. Despite being arranged hierarchically in Figure 2, Figure 3, Figure 4, Figure 5, Figure 6 and Figure 7, mechanisms, and pathways of PBDEs toxicity are intertwined. Some studies have shown the opposite results. The potential explanations could include the low-dose effects, nonmonotonic effects, compensate mechanisms and exposure route. The chemical formula C_12_H_(9–0)_Br_(1–10)_O has the sum of H and Br atoms fixed at 10. It is theorized that PBDEs comprise 209 congeners separated into ten homolog groups (ranging from mono- to decabromodiphenyl ethers). They belong to a similar structural group, i.e., they contain two aromatic bromine-containing rings (connected by an oxygen atom). Therefore, PBDEs are structurally similar and have analogous toxicological properties. Besides, different tissues may exhibit different toxic susceptibilities to the same compound. Given the complexity and diversity of signaling mechanisms responsible for PBDEs toxicity, it is difficult to consolidate them into particular classifications. Therefore, we summarize other mechanisms in the subtitle: “Others” and illustrate them in Table 1, Table 2, Table 3, Table 4, Table 5, Table 6, Table 7, Table 8 and Table 9.

Overall, we review the toxic effects of PBDEs on health, particularly focusing on liver toxicity, kidney toxicity, gut toxicity, thyroid toxicity, embryotoxicity, reproductive toxicity, neurotoxicity, immunotoxicity, heart toxicity and eye toxicity. We described the general mechanism of action, such as oxidative damage, apoptosis, disturbance of glucose and lipid metabolism, mitochondrial damage, intestinal microbiome disturbance, and hormonal interferences. This review highlights that PBDEs have a broad variety of toxic effects and mechanisms. Further studies regarding sub-cellular localization, actual environmental exposure levels, target tissue doses, tissue specificity, dose-effect relationship, and nuanced mechanisms are required. There is an ongoing need to further elucidate the specific mechanisms, assimilate new research outcomes into the existing framework of PBDE toxicity and expand existing measures to mitigate potential health effects associated with PBDE exposures.

## 12. Future Directions

Our review offers a summary of the toxic effects and mechanisms of PBDEs. Special attention should be paid to the persistence, bioaccumulation, and hazards of PBDEs. More work is needed to further explore the molecular mechanisms that explain the association between health outcomes and PBDEs. Understanding the adverse health effects and potential mechanisms of PBDEs helps in developing strategies to minimize exposure and mitigate potential health risks.

## Figures and Tables

**Figure 1 ijms-24-13487-f001:**
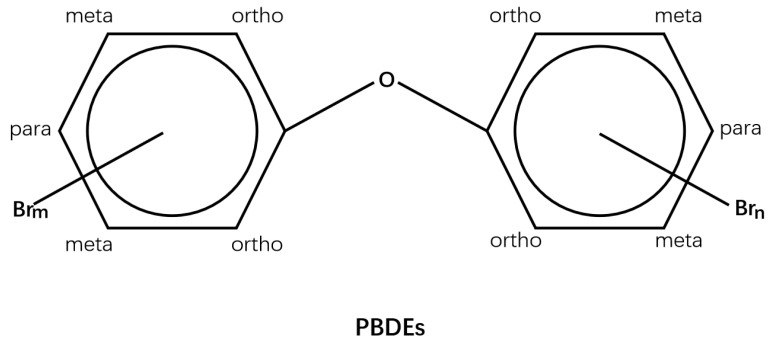
Structures of polybrominated diphenyl ethers (PBDEs). PBDEs consist of two benzene rings connected by an oxygen atom. A total of 209 PBDE congeners named according to the number of bromine atoms and their position (i.e., ortho-, meta-, and para-substitution) are included in PBDEs. m + n = 1–10.

**Figure 2 ijms-24-13487-f002:**
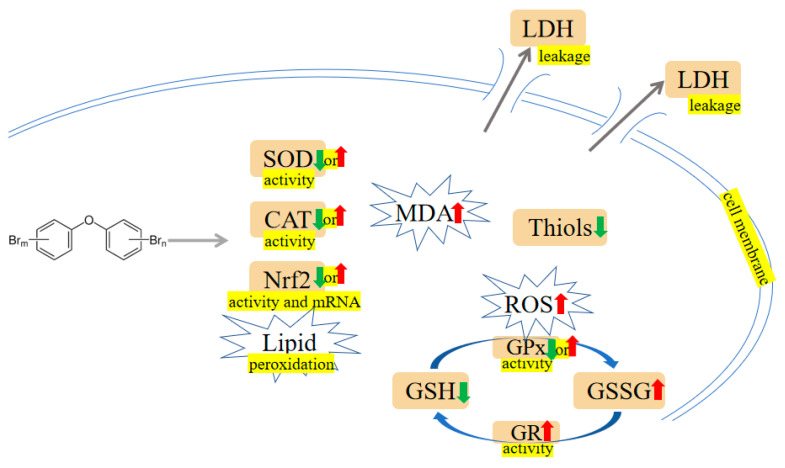
PBDEs-induced toxicity is associated with oxidative damage. PBDEs exposure can alter antioxidant enzyme activities, generate reactive oxygen species (ROS), increase malondialdehyde (MDA), and induce lactate dehydrogenase (LDH) leakage. Arrows indicate up (red colour), increased; down (green colour), decreased; up or down, increased or decreased (opposite research results exist). The yellow highlighted text is an explanation of the figure.

**Figure 3 ijms-24-13487-f003:**
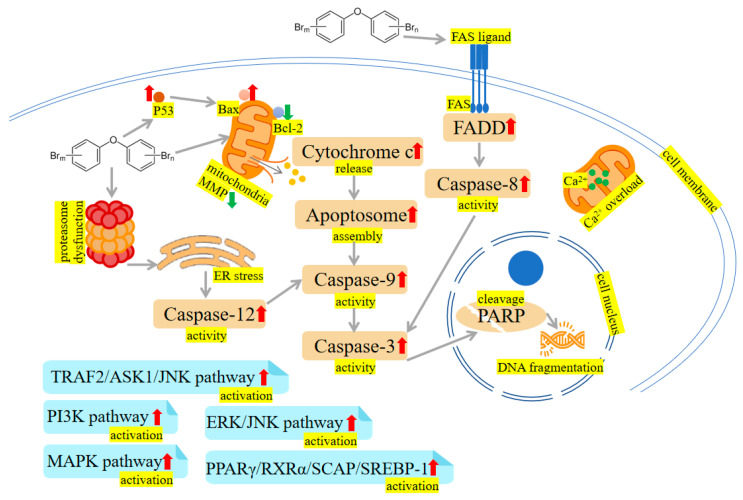
PBDEs-induced toxicity is associated with apoptosis. PBDEs exposure can reduce mitochondrial membrane potential (MMP), increase caspase activities, disrupt calcium homeostasis, induce endoplasmic reticulum (ER) stress and damage DNA. TNF receptor-associated factor 2 (TRAF2)/apoptosis signal-regulating kinase 1 (ASK1)/c-Jun N-terminal kinase (JNK) pathway, phosphoinositide-3-kinase (PI3K) pathway, extracellular signal-regulated kinase (ERK)/c-Jun N-terminal kinase (JNK) pathway, mitogen-activated protein kinase (MAPK) pathway and peroxisome proliferator-activated receptor γ (PPARγ)/retinoid X receptor α (RXRα)/sterol regulatory element-binding protein cleavage-activating protein (SCAP)/sterol regulatory element-binding protein-1 (SREBP-1) pathway are activated by PBDEs. Arrows indicate up (red colour), increased; down (green colour), decreased. The yellow highlighted text is an explanation of the figure.

**Figure 4 ijms-24-13487-f004:**
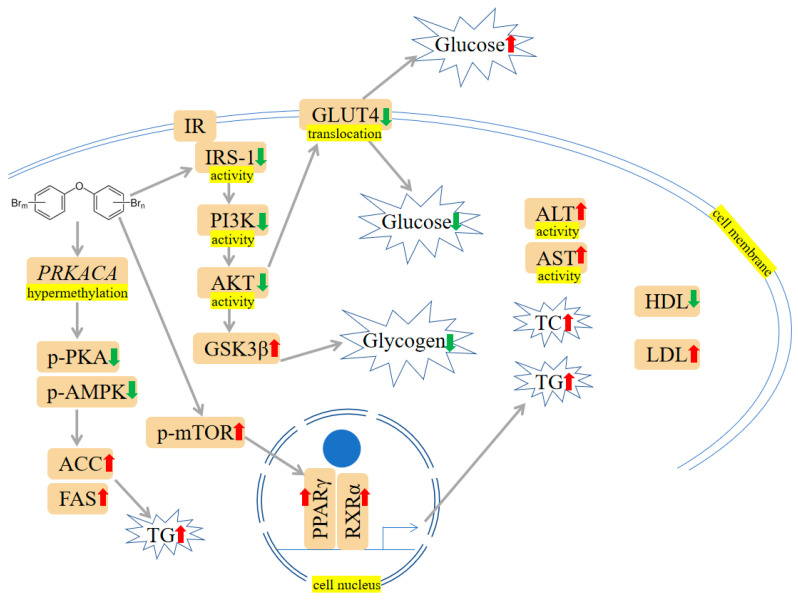
PBDEs-induced toxicity is associated with disturbances in glucose and lipid metabolism. PBDEs could increase glucose, total cholesterol (TC), triglyceride (TG), aspartate transaminase (ALT) activity, and aspartate transaminase (AST) activity. The PI3K/protein kinase B (AKT)/Glucose transporter type 4 (GLUT4) pathway is inhibited, and the mammalian target of rapamycin (mTOR)/PPARγ/RXRα pathway is elevated. Arrows indicate up (red colour), increased; down (green colour), decreased. The yellow highlighted text is an explanation of the figure.

**Figure 5 ijms-24-13487-f005:**
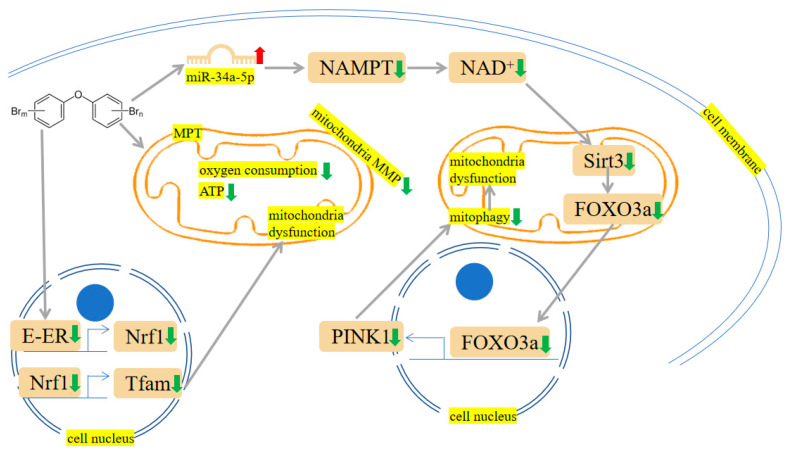
PBDEs-induced toxicity is associated with mitochondria damage. PBDEs exposure caused adenosine triphosphate (ATP) depletion, mitochondrial permeability transition (MPT) induction and mitochondria dysfunction. Sirtuin 3 (Sirt3)/forkhead box O-3 α (FOXO3α)/PINK1 pathway is suppressed by PBDE-209. Arrows indicate up (red colour), increased; down (green colour), decreased. The yellow highlighted text is an explanation of the figure.

**Figure 6 ijms-24-13487-f006:**
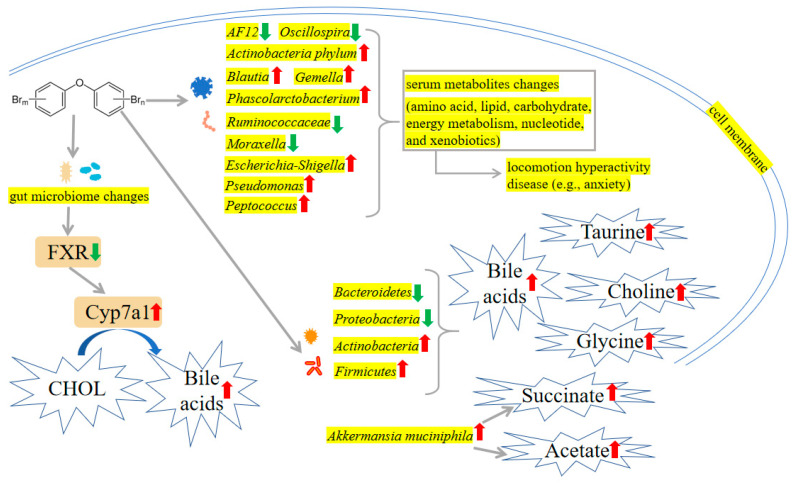
PBDEs-induced toxicity is associated with intestinal microbiome disturbance. PBDEs exposure increased *Actinobacteria phylum*, *Blautia*, *Gemella*, *Phascolarctobacterium*, *Escherichia-Shigella*, *Pseudomonas*, *Peptococcus*, *Actinobacteria*, *Firmicutes*, and *Akkermansia muciniphila*. In addition, *AF12*, *Oscillospira*, *Ruminococcaceae*, *Moraxella*, *Bacteroidetes*, and *Proteobacteria* are decreased by PBDEs treatment. These intestinal microbiome changes may upregulate bile acids, taurine, choline, glycine, succinate, and acetate levels. Arrows indicate up (red colour), increased; down (green colour), decreased. The yellow highlighted text is an explanation of the figure.

**Figure 7 ijms-24-13487-f007:**
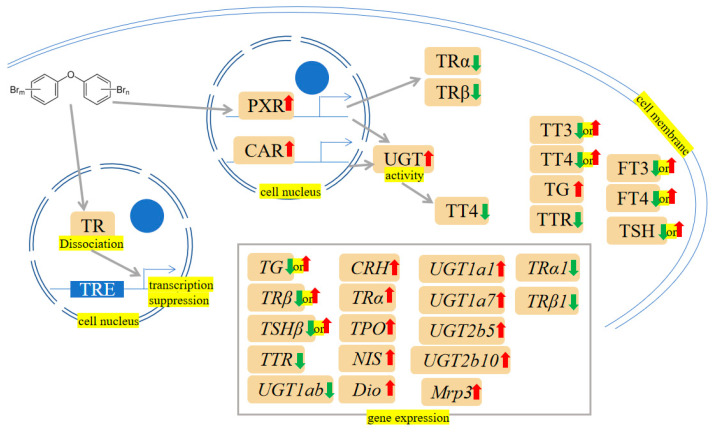
PBDEs-induced toxicity is associated with thyroid dysfunction. PBDEs exposure causes thyroid receptor (TR) dissociation from the TH response element (TRE), and the subsequently related gene expression may be affected. Constitutive androstane receptor (CAR)/pregnane X receptor (PXR) pathway may also be activated by PBDEs. Arrows indicate up (red colour), increased; down (green colour), decreased; up or down, increased or decreased (opposite research results exist). The yellow highlighted text is an explanation of the figure.

**Table 1 ijms-24-13487-t001:** Effects and mechanisms of liver toxicity induced by PBDEs.

Treatments	Effects and Mechanisms	References
PBDE-47 or -153, zebrafish	CAT activity↑, SOD activity↑, *Caspase-3*↑, *P53*↑, *Bcl-2*↓	[26]
PBDE-47 or -32, HepG2 cells, trout liver cells	Cell viability↓, ROS↑, apoptosis, DNA damage, mitochondrial impairment	[42,43,45]
PBDE-47, HSCs	ROS↑, lipid peroxidation, MMP↓	[44]
PBDE-47, -99, -209, HepG2 cells	*ERα*↑, *PPARα*↑, intracellular lipid accumulation	[27]
PBDE-47, CAR and PXR null mice	*CYP2B6*↑, CYP2B6↑	[85]
PBDE-47 or -99, isolated Wistar rat liver mitochondria	oxygen consumption↓, mitochondrial swelling, calcium release, ATP↓	[73]
PBDE-47, CD-1 mice, ICR mice, C57 BL/6 mice	Proteasome dysfunction, TRAF2/ASK1/JNK pathway↑, NAD^+^ depletion, Sirt1↓, inflammation↑, abnormal insulin secretion, *miR-34a-5p*↑, Sirt3/FOXO3α/PINK1 pathway↓, mitochondrial dysfunction	[1,46,47,57,72]
PBDE-99, SD rats, HepG2 cells	SOD activity↑, CAT activity↓, GSSG↑, GSH↓, Caspase-3 activity↑, Caspase-9 activity↑, apoptosis	[48,49]
PBE-99, C57BL/6 mice	Inflammation, acetate↑, succinate↑	[83]
PBDE-209, C57BL/6 mice, ICR mice, LO2 cells	ER stress↑, mitochondrial Ca^2+^ overload, apoptosis, ROS↑, PI3K/AKT/GLUT4 pathway↓, mTOR/PPARγ/RXRα pathway↑, Glucose↑, TG↑, HDL↓, liver and adipose structures damage	[50,68,69]
PBDE-209, SD rats	Hyperglycemia, GSH↓, SOD activity↓, liver weight↑, liver/body weight ratio↑, serum total bilirubin and indirect bilirubin↑, oxidative stress, *PXR*↓, *CAR*↓, CYP3A↓	[70,87]
PBDE-209, IR-BRL cells	TC↑, TG↑, AST activity↑, ALT activity↑, MDA↑, IRS-1/PI3K/AKT/GSK-3β pathway↓, IRS-1/GLUT4↓	[71]
PBDE quinone, LO2 cells	ER stress↑, autophagy-lysosomal system↑, ROS↑	[51]
PBDE-209, SD rats, LO2 cells	*PRKACA-1* hypermethylation, TG↑, Glucose↑, PI3K/AKT/GLUT4 pathway↓, mTOR/PPARγ/RXRα pathway↑	[62,68]
PBDE-209, HepG2 cells, isolated mitochondria	Mitochondrial Ca^2+^ overload, apoptosis, ROS↑, LDH leakage, matrix swelling, ATP↓, cell viability↓	[52,53,75]
PBDE-209, *Carassius auratus*	GR activity↑, GSH↓	[54]
PBDEs in e-waste site, kingfisher (*Alcedo atthis*)	MDA↑, ROS↑, CAT activity↓, SOD activity↓	[55]
PBDE-47, marine medaka (*Oryzias melastigma*)	PI3K pathway activity↑, MAPK pathway activity↑	[56]
PBDE-71, Wistar rats	Glucose:insulin ratio↑	[59,60]
PBDE-71, C57BL/6 mice	Glucose intolerance, fasting hyperglycemia, retarded glucose clearance, diminished thermogenic brown adipose tissue mass	[61]
PBDE-153, isolated rat liver mitochondria	MMP↓, ATP↓, ROS↑	[74]
PBDE-47, Wistar Han rats, indirect exposure	Centrilobular hypertrophy, fatty change, *cytochrome p450*↑, *Nrf2*↑, lipid↑, oncogenes change, epigenetic change	[76]
PBDE-99, SD rats, indirect exposure	PIP3K/AKT pathway↓	[77]
PBDE-47 and high-fat diet, HepG2 cells, C57BL/6J mice, combined exposure	*CPT1α*↓, fatty acid oxidation↓, microsomal triglyceride transfer protein↓, sterol regulatory element-binding protein 1↑, stearoyl-CoA desaturase 1↑, fatty acid synthase↑, lipid deposition, NAFLD, MDA↑, ROS↑, lipid accumulation	[19,78]
PBDE-47 and nanoplastics, zebrafish, combined exposure	Darker/browner liver colour, atrophied liver, liver degeneration or necrosis, *gpx1a*↓, CYP1A1↑, mortality↑, voluntary movements↑, hatching rate↑, heart rate↓	[79]
PBDE-47 and microplastics, grouper (*Epinephelus moara*), combined exposure	PPAR-related genes↑, IL-17-related genes↓	[18]
PBDE-71, hepatocytes derived from embryonic chickens	*TTR*↓, *THRSP14-α*↓, *FABP*↓	[80]
PBDE-71, B6C3F1/N mice	*Tbx3* hypomethylation	[81]
PBDE-47, PBDE-153 and PBDE-154 (alone or in combination), primary Atlantic salmon hepatocytes	*VTG*↑, *ZP3*↑	[82]

↑ represents upregulation, ↓ represents downregulation.

**Table 2 ijms-24-13487-t002:** Effects and mechanisms of kidney toxicity induced by PBDEs.

Treatments	Effects and Mechanisms	References
PBDE-99, SD rats	CAT activity↓, GSSG/GSH↑	[48]
PBDE-47, HEK293 cells	Cell apoptosis, ROS↑, *Bax*↑, *Bad*↑, *Bcl-2*↑, *Hrk*↑, ethanol↑, GSH↓, creatine↓, aspartate↓, UDP-glucose↓, NAD^+^↓	[88]
PBDE-47, CIK cells	CAT activity↓, SOD activity↓, GPx activity↓, T-AOC↓, Bax↑, Cytochrome C↑, Caspase-3↑	[89]
PBDE-47, C57BL/6 mice	Cytochrome c release, caspase activation, PARP cleavage, CAT activity↓, SOD activity↓, GPx activity↓, Nrf2 activity↓, ROS↑, NF-κB↑, ACR↑, NLRP3↑, CXCR4/TXNIP/NLRP3↑	[90,96]
PBDE-209, broilers	Swelling and granular degeneration of the renal tubular epithelium, atrophy and necrosis of glomeruli, MDA↑, GSH-Px↓, GSH↓, SOD↓, Bax/Bcl-2 ratio↑, p-ERK1/2↑, p-JNK1/2↑, Bax↑, Cytochrome c↑, Caspase-3↑	[91]
PBDE-209, Wistar rats	GSH↑, TBARS↑, -SH↓	[92]
PBDE-47 and Cd, HKC cells, combined exposure	Cell rounding, cell swelling, renal tubular epithelial cell damage, LDH release, NLRP3↑, cleaved Caspase-1↑, cleaved GSDMD↑, mitochondrial dysfunction, pyroptosis	[93]
PBDE-47, PK 15 cells	Mitochondrial fusion and fission↓, MMP↓, ROS↑, ATP↓, cellular disintegration	[94]
PBDE-47, CIK cells	AMPK-Sirt1-PGC-1α pathway↓, cytoplasmic Ca^2+^↑, *miR-140-5p* miRNA↓, TLR4↑, NF-κB↑	[89,95]

↑ represents upregulation, ↓ represents downregulation.

**Table 3 ijms-24-13487-t003:** Effects and mechanisms of gut toxicity induced by PBDEs.

Treatments	Effects and Mechanisms	References
PBDE-71, zebrafish	Disruption of epithelial barrier integrity, inflammatory response, and anti-oxidant capacity	[97]
PBDE-209, Caco-2 cells	*Nrf2*↓, *FAS*↑, *CYP1A1*↑	[98]
PBDE-47, SD rats	*AF12*↓, *Oscillospira*↓, *Actinobacteria phylum*↑, *Blautia*↑, *Gemella*↑, *Phascolarctobacterium*↑	[99]
PBDE-47, CD-1 mice, ICR mice	Fecal and liver bile acids↑, *CYP7A1*↑, FXR signaling↓, microbial diversity↓, microbial compositional alterations, worsen HFD-induced obesity, hepatic steatosis, and injury	[101,102,103]
PBDE-47 and -99, C57BL/6 mice	Unconjugated bile acids↑, *Akkermansia muciniphila*↑, acetate↑, succinate↑, Ntcp↓, Oatp1b2↓	[83,104]

↑ represents upregulation, ↓ represents downregulation.

**Table 4 ijms-24-13487-t004:** Effects and mechanisms of thyroid toxicity induced by PBDEs.

Treatments	Effects and Mechanisms	References
PBDE-47, HepG2 cells	*TRα1*↓, TRα1↓, *TRβ1*↓, TRβ1↓	[106]
PBDE-47, zebrafish	TR↓, head trunk angle↓, otic vesicle length↑, eye pigmentation↓, developmental delays	[107]
PBDE-47, zebrafish	*TTR*↓, *TG*↓, *TRβ*↓, *TSHβ*↓, *NIS*↑, *TPO*↑, *TRα*↑	[108]
PBDE-209, zebrafish	*CRH*↑, *TSHβ*↑, *NIS*↑, *TG*↑, *Dio1*↑, *Dio2*↑, *TRα*↑, *TRβ*↑, *TTR*↓, *UGT1ab*↓	[109]
PBDE-71, SD rats	Plasma T4↓, liver vitamin A↓, body weight↓, T3 (F1 Female)↑, T4 (F1 Female)↑, thyroid gland weight↑, *osteopontin*↑	[110,144]
PBDE-71, C57BL/6 mice	TT4↓, FT4↓	[111]
PBDE-209, workers	Positive relationship between serum PBDE-209 levels and total TH	[114]
PBDE-209, rainbow trout	OH-BDE metabolites negatively correlated with the plasma FT4 levels	[117]
PBDE-209, fathead minnows	TT4↓, TT3↓	[119]
PBDE-47, C57 BL/6 mice	TT4↓, *Ugt1a1*↑, *Ugt1a7*↑, *Ugt2b5*↑, *CYP2B10*↑, *Mrp3*↑	[121]
PBDE-47, SD rats	Apoptosis, Caspase-3 activation, PARP cleavage, DNA fragmentation↑, GRP78↑, ATF4↑, CHOP↑, p62 accumulation, ER stress, defective autophagy	[122,123,124]
PBDE-28 or -47, human	TT4↑, FT4↑, TT3↑, FT3↑	[135]
PBDEs, human	Placental T4↓ (PBDE-99, or -100)	[136]
PBDEs, human	TT4↓ (PBDE-99, -154 or -196), TT3↓ (PBDE-47, -99, -100, -197, -203 or -207)	[137]
PBDEs, human	TT3↑ (PBDE-47, -66 or 85), TT4↑ (PBDE-66, -85, 153 or -154), TT4↓ (PBDE-209)	[138]
PBDEs, human	TT4↓ (PBDE-28, -47, -99, -100 or -153)	[140]
PBDEs, human	TT3↓ (PBDE-154), FT3↓ (PBDE-153, -183), T4/T3 ratio↑ (PBDE-100)	[142]
PBDE-47 and PS-NPs, zebrafish, combined exposure	Deformity in pericardial edema, yolk sac edema, the curvature of the tail, *TSHβ*↑, *TG*↑, *Dio2*↑, *TRβ*↑	[146]
PBDE-209, JEG-3 cells	Dio3↓, *Dio3*↓, *has-miR-668-3p*↓, *has-miR409-3p*↓	[147]

↑ represents upregulation, ↓ represents downregulation.

**Table 5 ijms-24-13487-t005:** Effects and mechanisms of embryotoxicity induced by PBDEs.

Treatments	Effects and Mechanisms	References
PBDE-47, ICR mice	MAPK signaling↑, changed placental function, low birth weight, stillbirth rate↑, plasma testosterone↓, progesterone↓, growth hormone↓, compromised fetal development	[148]
PBDE-47, zebrafish	Embryonic development abnormalities, ROS↑, JNK activity↑	[149,150]
PBDE-209 or -47, hESCs, mESCs	ROS↑, OCT4↓, apoptosis, *OCT4*↓, *SOX2*↓, *NANOG*↓	[151,152]
PBDE-209 and nSiO_2_, zebrafish, combined exposure	Postpone hatching, heartbeat↓, arrhythmia↑, malformation↑	[153]
PBDEs, SD rats, zebrafish, *common terns*, *kestrels*	Soft tissue syndactyly or malposition of the distal phalanges and decreased ossification of the sixth sternebra (rats), embryo yolk sac, pericardial edema, spine deformation, neurobehavioral abnormalities, and blood vessels damage (zebrafish), shorter humerus length and reduced total thyroid weight (kestrels)	[154,155,156,157,158]
PBDE-47, human	Placental Dio3 methylation (female infants)↑	[159]
PBDE-47, ICR mice	Adverse pregnancy results, VEGF-A↓, placental angiogenesis↓	[160]
PBDE-47, CTB	Cell viability↓, Global CpG methylation↑	[161]
PBDEs, human	Fetal growth retardation (PBDE-206, PBDE-17-190, PBDE-196-209), aberrant methylation of *HSD11B2* and *IGF2*	[162]
PBDE-209, C57 mice	Fetal growth and development↓, TCA cycle↓, accelerated lipolysis, TH↓	[163]
PBDE-209, SD rats	ET-1↑, iNOS↑, eNOS↓, birth weight of the newborns↓	[164]
PBDE-209, C57BL/6 mice	Placental vascular growth↓, placental cell death, GRP78↑, PERK signaling↑	[165]
PBDEs, zebrafish	Yolk and pericardial edema, tail, and head malformation, reduced and extremely reduced heartbeat rate, blood stasis and spinal curvature, cardiac edema, damage of eye structure and hydrocephaly, liver vacuolization (PBDE-47, -99, -209), *TSHβ*↑, *TTR*↑, *Tbg*↑, *Dio1*↑ (PBDE-47, -99), *Dio1*↑ (PBDE-209)	[166]

↑ represents upregulation, ↓ represents downregulation.

**Table 6 ijms-24-13487-t006:** Effects and mechanisms of reproductive toxicity induced by PBDEs.

Treatments	Effects and Mechanisms	References
PBDE-209, ICR mice, CD-1 mice, Parkes strain mice, Balb/c mice, Sertoli cells	Oxidative stress, testosterone↓, DNA damage, ATM/Chk2↑, ATR/Chk1↑, DNA-PKcs/XRCC4/DNA ligase IV pathways↑, impaired germ cell proliferation, germ cell apoptosis↑, cx43↓, p27Kip1↓, ER signaling↑, impaired blood-testis barrier, sperm quality↓, arrested meiotic prophase I, testicular size↓, spermatogenesis↓	[167,168,171,172,173,174,175,201,207,208,209,210]
PBDE-209, SD rats	Mitochondrial function↓, telomeres length↓, telomerase activity↓, PPARγ/RXRα/SCAP/SREBP-1↑, cell apoptosis, anogenital distance↓, abnormal sperm morphology, blood-testis barrier ultrastructure damage, tight junctions damage, ectoplasmic specialization structures with broken tight junctions, actin microfilaments↓	[169,170,189,200]
PBDE-47, *Brachionus plicatilis*, SD rats	Reproductive rate↓, intrinsic increase rate↓, impaired ultrastructure of the ovary, ROS↑	[176,177]
PBDE-47, marine medaka (*Oryzias melastigma*), manila clam *Ruditapes philippinarum*	Histone variants↓, parvalbumins↓, spermatogenesis↓, infertility, vitellogenins↑, apolipoprotein A-I↑, reproductive dysfunction	[211,212]
PBDE-47, mice, SD rats	Mitochondrial disruption, aberrant distribution, MMP↓, apoptosis	[178,179]
PBDE-47, Wistar rats, SD rats	Histone-protamine exchange↓, aberrant sperm epigenome, DMRs↑, autophagy↑, ovary damage, testosterone signaling disruption, AR antagonism, mTORC1 signaling↑, replacement of thyroid hormone from transporting proteins, cAMP↑	[187,188,213,217]
PBDE-47, GC2 cells, ICR mice	Cell viability↓, condensation of nuclear, vacuolated mitochondria, Atp5b↓, Uqcrc↓, MMP↓, apoptosis, ER-Nrf1-Tfam-mitochondria pathway disturbance, mitochondria function↓, spermatogenesis↓, germ cells damage	[180,196]
PBDE-47, *Fathead Minnows* (*Pimephales promelas*)	Clutch size↓, fecundity↓	[214]
PBDE-47, BeWo Cells	Mitochondria function↓, cholesterol transport↓, progesterone synthesis↓	[197]
PBDE-47, ICR mice, GC-2 cells	Spermatogenesis damage, SETD8/H4K20me1-linked histone methylation disturbance, meiosis initiation↓, cell cycle progression↓, male reproductive toxicity	[190,191]
PBDE-71, zebrafish, male American kestrels (*Falco sparverius*)	Malformation, percentage of male↑, male courtship behaviors↓	[215,216]
PBDE-99, Leydig cells	ROS↑, ERK1/2 pathway↑, ubiquitination degradation pathway↓, apoptosis	[181]
PBDE-99, Wistar rats	Adverse ultrastructural changes of mitochondria	[198]
PBDEs, Luteal cells	Malfunction of the corpus luteum, initiating apoptosis (PBDE-47, -99, -100)	[182]
PBDE-3, SD rats	ROS↑, serum testosterone↓, Leydig cell size↓, p-ERK1/2↓, p-AKT↓, p-AMPK↓	[183,184]
PBDE-3 or -47, *Caenorhabditis elegans*	Life spans↓, fecundity↓, delayed egg-laying, ROS↑, DNA damage	[185,186]
PBDE-47 and high-fat diet, SD rats, combined exposure	Exacerbated the damage to the seminiferous epithelia, testosterone↓, spermatozoa↓, DAX-1↑, StAR↓, 3β-HSD↓	[199]

↑ represents upregulation, ↓ represents downregulation.

**Table 7 ijms-24-13487-t007:** Effects and mechanisms of neurotoxicity induced by PBDEs.

Treatments	Effects and Mechanisms	References
PBDE-209, Neuro-2a cells, HT-22 cells	FAS↑, FADD↑, Caspase-8↑, Caspase-3↑, apoptosis, *PDEs*↑, Bcl-2/Bax↓	[219,220]
PBDE-47 or -209, Neuro-2a cells	*P53*↑, *P21*↑, *cycline D1*↓, *CDK2*↓, *Nrf2*↑, MMP↓, Cytochrome c release↑, Caspase-9↑, Caspase-3↑, ROS↑, MDA↑, GSSG/GSH ratio↑	[265,266]
PBDE-209, SD rats, Wistar rats	LC3-Ⅱ↑, Beclin-1↑, P62↓, cleaved caspase-3↑, cleaved PARP↑, Bcl-2↓, neurons death, synaptic plasticity↓	[221,267]
PBDE-153, SD rats	Learning ability↓, spontaneous activity↓, neuron apoptosis	[222]
PBDE-47, C57BL/6 mice cerebellar granule neurons, hNPCs	Extracellular glutamate↑, ionotropic glutamate receptors↑, calcium↑, oxidative stress, cell death	[223,224]
PBDE-47, SD rats	Caspase3↑, Caspase12↑, Cytochrome C↑, *Caspase3*↑, *Caspase12*↑, *Cytochrome C*↑, dendrites length↓, spines density↓, the behavior of autism, motor defect, impaired dopaminergic system, α-synuclein aggregation, ubiquitination↓, autophagy↓, PD risk↑, hyperactivity and anxiety-like behavior, *Ruminococcaceae* and *Moraxella*↓, *Escherichia-Shigella*↑, *Pseudomonas* and *Peptococcus*↑	[225,233,236,237]
PBDE-99, cerebellar granular neurons, SD rats	*BDNF*↓, Bcl-2↓, learning difficulties, free radicals↑	[226,227,228]
PBDE-99, CD-1 Swiss mice	Altered thigmotaxis, time in the centre of the arena↑	[255]
PBDE-47, human	BDNF↓, risk of post-partum depression↑	[229]
PBDEs, NMRI mice	BDNF↓, Ca/CaMKII↑ (PBDE-209, -206, -203)	[230,231]
PBDEs, human	Depression symptoms (PBDE-47, -99)	[232]
PBDE-71, zebrafish	Serotonin↓, TH↓, dopamine transporter protein↓, hyperactivity (low concentrations), activity during the dark period↓ (high concentrations), calcium balance disruption, cholinergic function↓, locomotor activity↓	[234,235,258,259]
PBDE-47 and PS-NPs, zebrafish, combinde exposure	Accelerated voluntary movements, mortality↑, darker/browner liver colour, atrophied liver, *ache*↓, *chrn7*↓	[79]
PBDE-71, SD rats	Delayed effects on sensory reactivity↓, startle reactions↑	[257]
PBDEs, *Daphnia magna*	AChe activity↓ (PBDE-47, 6-OH-PBDE-47 and 6-MeO-PBDE-47)	[238]
PBDEs and PCB, zebrafish, combinde exposure	Neurobehavioral defects, mitochondrial function↓, lipid metabolism regulation↓	[239,240]
PBDE-209 and Pb, C57BL/6 mice, combined exposure	TNFα↑, IFNγ↑, IL-4↑, IL-6↑, IL-10↑, IL-17 A↑, neuronal cells↓, impaired learning behavior	[241]
PBDE-209, C57BL/6 mice, ICR mice	Glial cell activity↓, hippocampal growth↓, behavioral difficulties, dendritic branches↓, synaptic proteins↓, doublecortin↑, weight gain↓, litter size of maternal mice↓, TT3↑, TT4↑, FT3↑, FT4↑, *dio1*↓ (livers), *dio2*↓ (livers), *dio3*↓ (livers), *dio1*↑ (brains), *dio3*↓ (brains), dio3↓, calcium overload, impaired learning and memory	[254,268,269,270,271]
PBDE-209 and Pb, zebrafish, combined exposure	Locomotor dysfunction, ROS↑, lipid peroxidation, DNA damage, antioxidant system↓	[242]
PBDE-209 and nano-TiO_2_, zebrafish, combined exposure	Locomotion activity↓, *mbp*↓, *a1-tubulin*↓, *gap-43*↓	[243]
PBDE-99 and MeHg, SD rats, combined exposure	Developmental neurotoxic effects, impaired negative geotaxis reflexes, impaired motor coordination	[244]
PBDE-99 or PBDE-47, NMRI mice, cerebral cortex cells, Wistar rats, human glial cells	Development of the brain↓, spontaneous behavior disturbance, Gap 43↑, Dio2 activity↓, neurodevelopmental deficits	[246,247,248,272]
PBDE-47, zebrafish	Spontaneous coil activity↑, locomotion↓, touch response↓, swimming speed↓, axonal growth↓, motor deficits, Hedgehog signaling↓	[249,250,261,262]
PBDE-47 or -209, hNSC	NOTCH, GSK3β and T3 signaling interference	[251]
PBDE-47 or -49, SD rats, primary hippocampal cell cultures	Neuronal polarization delay, axonal outgrowth↓, RyR activity↑	[264]

↑ represents upregulation, ↓ represents downregulation.

**Table 8 ijms-24-13487-t008:** Effects and mechanisms of immunotoxicity induced by PBDEs.

Treatments	Effects and Mechanisms	References
PBDE-47, SD rats	NETs formation, ROS↑	[274]
PBDE-47, *Ruditapes philippinarum* hemocytes, *Mytilus edulis*	Phagocytic ability↓, bacteriolytic activity↓, ROS↑, alteration of MAPKs pathways, lysosomal membrane damage	[275,276]
PBDE-47 or- 209, Kunming mice	ROS↑, GSH↓, macrophage accessory cell function↓	[277]
PBDEs, harbour seal immune cells	ROS↑, thiols↓ (PBDE-47, -99, -153)	[278]
PBDE-47, THP-1 macrophage-like cells, spleen-derived lymphocytes, BALB/c mice, human	Perturb the innate immune response, disrupt the secretion of proinflammatory cytokines (IL-6 and TNF-α) and interfere with basophil activation	[279,281,282,283,284]
PBDE-47, human PBMC	Higher response to LPS	[285]
PBDE-47, fathead minnows (*Pimephales promelas*), rainbow trout (*Oncorhynchus mykiss*)	Resistance to the pathogen↓, survival rate↓, head kidney impairment, immune factors↓, respiratory burst activity↑, immune-related genes↓, impaired immune organs	[286,287]
PBDE-209, C57BL/6 mice, Balb/c mice, SD rats	Proliferative effects↓, production of cytokines↓, atrophying immune organs, humoral and cellular immunity changes	[288,289,290,291]
PBDE-209, Broiler chicks	Damaged and necrotic lymphocytes, lymphoid cells in the thymic gland↓	[292]
PBDEs, pantropical spotted dolphin	Inflammatory cytokine↑, PGE2↑, cAMP↑, *COX-2*↑ (PBDE-47, -100, -209)	[294]
PBDE-47, CBZ and CdCl_2_, gilthead sea bream (*Sparus aurata* L.), combined exposure	Dysregulation of pro-inflammatory factors and humoral response in the serum or skin mucus	[295]

↑ represents upregulation, ↓ represents downregulation.

**Table 9 ijms-24-13487-t009:** Effects and mechanisms of toxicity (diabetes, heart, eye and lung) induced by PBDEs.

Treatments	Effects and Mechanisms	References
PBDE-47, SD rats	Risk of diabetes prevalence↑, hyperglycemia, scattered microvesicular steatosis	[296]
PBDEs, human	Glucose homeostasis disturbance, gestational diabetes mellitus↑	[297]
PBDE-209 and high-fat diet, C57BL/6 mice	Blood glucose↑, insulin signaling pathway↑, *GLUT4*↓, *TRα*↓, *AR*↓, *Insr*↓	[298]
PBDE-209, SD rats	Impaired morphology and ultrastructure of the heart and abdominal aorta, serum creatine kinase↑, LDH↑, IL-1β↑, IL-6↑, IL-10↑, TNFα↑, endothelial dysfunction, cardiovascular injury	[299]
PBDE-209, human umbilical vein endothelial cells	ROS↑, IRE1α/AKT/mTOR signaling pathway↑, autophagy↑, apoptosis	[300]
PBDE-71, zebrafish	Area of inner plexiform layer↓, inner nuclear layer↑, density of ganglion cells↓, hyperactive responses, retinal and retinyl ester content↑, *raldh2*↑, *rdh1*↓, *crabp1a*↓, *crabp2a*↓, *raraa*↓ *zfrho*↑, *zfuv*↑, *zfred*↑, *zfblue*↑, *and zfgr1*↑	[301,304,305]
PBDE-47, hESC-ROs, zebrafish	Thickness and area of the neural retina↓, cell proliferation↓, cell apoptosis, aberrant differentiation, abnormal eye morphogenesis	[302,303]
PBDE-209, human lung epithelial cell	LDH leakage↑, cell viability↓, IL-6↑, IL-8↑, *IL-6*↑, *IL-8*↑	[306]
PBDE-47, -99 or -209, A549 cells, pNHBE cells, human bronchial epithelial cells	Inflammation, oxidation stress, barrier integrity↓, uncontrolled production of mucous, alterations in physics and biochemical properties of airway fluids, NOX-4↑, ROS↑, DNA damage and repair processes↑	[307,308,309]
PBDEs, human lung epithelial cells	Membrane disruption, LDH leakage↑, oxidation stress, MMP↓, ROS↑	[310]

↑ represents upregulation, ↓ represents downregulation.

## Data Availability

Not applicable.

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
