# Peer review of "Toxic Effects and Mechanisms of Polybrominated Diphenyl Ethers"

_ijms, 2023, doi:10.3390/ijms241713487_

Round 1

Reviewer 1 Report

The manuscript is a well-structured and complete review on the toxicity and toxicodynamic of PBDEs

The volume of cited literature is extensive, updated, relevant, and devoid of excessive self-citations.

The summarizing table is a very useful tool for a quick reference on each congener of PBDEs group.

Figures are well-organized and a good summary of toxicity mechanisms of PBDEs.

The only note is represented by the conclusions, which look more like an introduction, and which would be better moved to the appropriate paragraph. in addition to a brief summary of the problems exposed, the reader could be interested in some input for possible mitigation measures of the problem represented by the diffusion in the environment and biota of this class of flame retardants, an aspect that has not been described in the text. 

Reviewer 2 Report

Review of the manuscript entitled: Toxic Effects and Mechanisms of Polybrominated Diphenyl Ethers. The manuscript is interesting, but some corrections are needed. The manuscript is very well prepared, and I find no serious remarks. Moreover, in my opinion the paper is prepared at a very high level.

1.      In abstract and introduction clear aim of the manuscript should be added e.g. "The aim of the present study was to ...".

2.      References should be added to: 52-53, 82, 269, lines.

3.      Abbreviation should be added line 92 (H2DCFDA)

4.      Chapter 2.2 “Disturbance of glucose and lipid metabolism.” In my opinion PPARgamma is crucial. If you can add more information about this receptor.

5.      It seems to me that the abbreviations are explained several times please check the whole manuscript carefully

6.      If possible, figures and tables should be placed between large sections and not at the end of the manuscript.
